# Epigenetic modulation of inflammation and synaptic plasticity promotes resilience against stress in mice

Jun Wang[1,2], Georgia E. Hodes[3], Hongxing Zhang[4], Song Zhang[4], Wei Zhao[1], Sam A. Golden [3], Weina Bi[1], Caroline Menard [3], Veronika Kana[5], Marylene Leboeuf[5], Marc Xie[1], Dana Bregman[3], Madeline L. Pfau[3], Meghan E. Flanigan[3], Adelaida Esteban-Fernández[1], Shrishailam Yemul[1], Ali Sharma[1], Lap Ho[1], Richard Dixon [6], Miriam Merad[5], Ming-Hu Han[3,4], Scott J. Russo [3] & Giulio M. Pasinetti [1,2]

Major depressive disorder is associated with abnormalities in the brain and the immune system. Chronic stress in animals showed that epigenetic and inflammatory mechanisms play important roles in mediating resilience and susceptibility to depression. Here, through a high-throughput screening, we identify two phytochemicals, dihydrocaffeic acid (DHCA) and malvidin-3′-O-glucoside (Mal-gluc) that are effective in promoting resilience against stress by modulating brain synaptic plasticity and peripheral inflammation. DHCA/Mal-gluc also significantly reduces depression-like phenotypes in a mouse model of increased systemic inflammation induced by transplantation of hematopoietic progenitor cells from stress-susceptible mice. DHCA reduces pro-inflammatory interleukin 6 (IL-6) generations by inhibiting DNA methylation at the CpG-rich IL-6 sequences introns 1 and 3, while Mal-gluc modulates synaptic plasticity by increasing histone acetylation of the regulatory sequences of the Rac1 gene. Peripheral inflammation and synaptic maladaptation are in line with newly hypothesized clinical intervention targets for depression that are not addressed by currently available antidepressants.

[1] Department of Neurology, Icahn School of Medicine at Mount Sinai, New York, NY 10029, USA. [2] Geriatric Research, Education and Clinical Center, James J. Peters Veterans Affairs Medical Center, Bronx, NY 10468, USA. [3] Department of Neuroscience, Icahn School of Medicine at Mount Sinai, New York, NY 10029, USA. [4] Department of Pharmacological Sciences, Icahn School of Medicine at Mount Sinai, New York, NY 10029, USA. [5] Department of Oncological Sciences, Tisch Cancer Institute and Immunology Institute, Icahn School of Medicine at Mount Sinai, New York, NY 10029, USA. [6] Department of Biological Science, University of North Texas, Denton, TX 76203, USA. Correspondence and requests for materials should be addressed to G.M.P. (email: giulio.pasinetti@mssm.edu)

Currently available treatments for major depressive disorder (MDD) mainly target neurochemical or neurobiological mechanisms[1]. Conventional pharmacological treatments produce temporary remission in <50% of patients[2,3]. Thus, there is an urgent need for a wider spectrum of novel therapeutics to target newly discovered underlying disease mechanisms.

Depression is associated with a multitude of pathological processes. Peripheral inflammation and synaptic abnormalities are thought to directly or indirectly induce brain functional abnormalities contributing to depression[4–8]. Patients with MDD display neuronal atrophy and reduced brain volume in regions including the prefrontal cortex, amygdala, hippocampus, ventral striatum, and thalamus. Post-mortem brain transcriptomic studies of MDD subjects identified alterations in the expression of genes important for synaptic functions[9,10]. Stress-induced abnormalities in synaptic remodeling are also observed in animal models of stress[11–14]. We previously found that both in humans and in rodents, chronic stress reduces RAS-related C3 botulinum toxin substrate 1 (Rac1) expression in the NAc and this downregulation is associated with repressive chromatin state surrounding its proximal promoter region[11]. Rac1 plays an important role in regulating dendritic spines and excitatory synapses[15–17]. In psychiatric disorders, Rac1-mediated spine morphological changes may contribute to pathological structural and functional changes of the brain[18,19]. Pharmacological inhibition of histone deacetylases (HDACs) not only normalizes Rac1 transcription but also rescues depression-like behavior[11]. These studies suggest Rac1-mediated synaptic remodeling in the NAc, and associated epigenetic changes, can be potential targets for depressive disorders.

In addition to the central nervous system (CNS), the contribution of peripheral inflammation to depression has received increasing attention. Many neuroimmune factors have been implicated in depressive disorders, however, studies in humans suggest that elevated expression of peripheral interleukin 6 (IL-6) is most consistently observed[20,21]. IL-6 is a proinflammatory cytokine and the mechanism(s) by which peripheral IL-6 may modulate depression phenotypes are under intense investigation. Current evidence suggests that IL-6 is capable of crossing the blood–brain barrier (BBB) and may alter neuroplasticity by acting directly on neurons or indirectly through modulation of microglia and/or other CNS immune cells[22]. This is supported by the observation that intracranial infusion of IL-6 increases depression-associated behavior[23]. These studies suggest that modulations of IL-6 and associated immune signaling pathways may provide novel therapeutic strategies to prevent and/or treat depression.

In spite of the important roles of inflammation and brain synaptic remodeling in the pathogenesis of depression, currently available antidepressants do not specifically address MDD-associated inflammation and synaptic maladaptation. Polyphenols have shown some efficacy in modulating aspects of depression; however, the mechanisms of action are largely unknown[24]. We recently identified a bioactive dietary polyphenol preparation (BDPP), composed of Concord grape juice, grape seed extract and trans-resveratrol, that is effective in protecting against diverse mechanisms associated with multiple neurological disorders[25–29]. Here we test the efficacy of BDPP and two novel phytochemicals derived from post-absorptive and microbiome metabolism of BDPP in attenuating depression by reversing stress-mediated brain synaptic maladaptation through modulation of peripheral IL-6 and Rac1 in the NAc. The study provides preclinical evidence supporting simultaneously targeting novel key disease mechanisms through epigenetic modification as a novel intervention for depression.

## Results

**Treatment with BDPP promotes resilience to social stress.** To test the efficacy of BDPP in promoting resilience in stress-mediated depression, we treated C57BL/6 male mice with BDPP or vehicle for 2 weeks prior to and throughout RSDS and then performed social avoidance/interaction (SI) testing (Fig. 1a). Treatment with BDPP significantly increased the proportion of mice resilient to stress as indicated by increased social interactions compared to the vehicle-treated animals (Fig. 1b, c). Overall, over 70% of mice receiving BDPP showed a resilient behavioral phenotype, whereas <40% were resilient in the vehicle control group.

**IL-6 and Rac1 modulate synaptic plasticity.** Evidence garnered from the RSDS model indicates the novel roles of IL-6 and Rac1 as contributing factors to depression phenotypes[11,30]. Recent studies demonstrate that increased glutamatergic transmission on ventral striatum medium spiny neurons (MSNs) mediates stress-induced susceptibility following RSDS[31,32]. We therefore

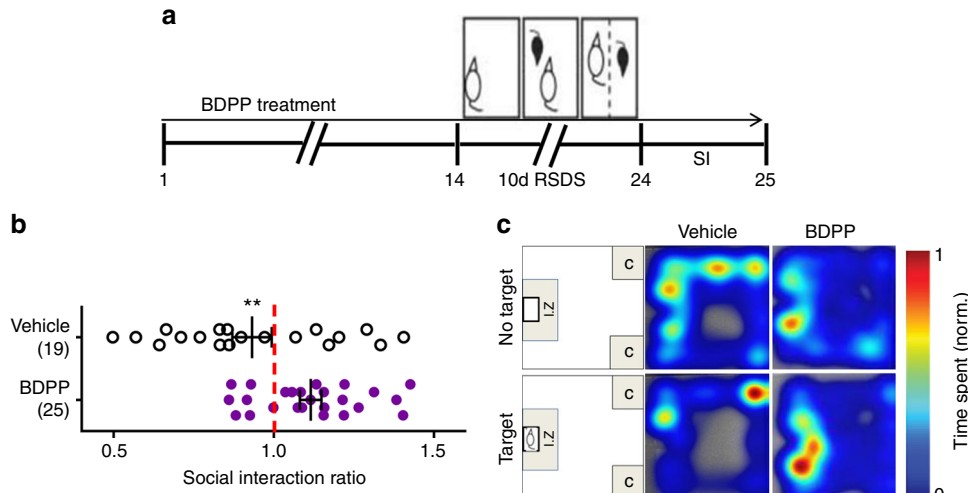

**Fig. 1** Oral administration of BDPP promotes resilience to RSDS. **a** Schematic design of the experiment. **b** Treatment with BDPP increases the proportion of mice showing a resilient phenotype, as measured by social interaction ratio (two-tailed unpaired t-test, $t_{42} = 2.786$, $P = 0.008$). **c** Representative heatmaps of social avoidance behavioral test. Graphs represents mean ± s.e.m., **$P < 0.01$

investigated the biological roles of IL-6 and Rac1 on glutamatergic synapses. To test the effect of peripheral IL-6 on synaptic plasticity in the NAc, we generated bone marrow (BM) transplanted chimeras using BM from wild type (WT) or IL-6 knockout mice (IL-6$^{-/-}$). Following transplantation and recovery, chimeras were exposed to RSDS and the density of PSD-95, a postsynaptic marker of excitatory synapses, was measured in the NAc. RSDS induced a robust increase of PSD-95 puncta in WT BM chimeras compared to unstressed WT or IL-6$^{-/-}$ BM chimeras, while no increase of PSD-95 puncta was observed in stressed IL-6$^{-/-}$ BM chimeras (Supplementary Fig. 1a). Whether stress-induced IL-6-mediated up-regulation of PSD95 in the NAc is specific to select cell types (e.g., Drd1 or Drd2 medium spiny neurons, or interneurons) needs further characterization. Measurements of circulating IL-6 revealed that IL-6 was significantly higher in WT chimeras compared to IL-6$^{-/-}$ BM chimeras following RSDS (Supplementary Fig. 1b). The level in stressed IL-6$^{-/-}$ BM chimeras was comparable to that of non-stressed WT chimera, suggesting leukocytes are the main source of circulating IL-6 and alternative sources may only make minor contributions to stress-induced IL-6 increase. Previous studies showed that Rac1 decreases excitatory spine density on MSNs and attenuates RSDS-induced susceptibility[11]. We found that exogenous expression of Rac1 in MSN-enriched primary cultures significantly reduced the expression of PSD-95 and vesicular glutamate transporter 2 (vGlut2) while having no effects on GABAergic vesicular GABA transporter (VGAT) (Supplementary Fig. 1c). Collectively, these data demonstrate that IL-6 and Rac1 can each modulate synaptic plasticity in neurons from NAc, supporting these mechanisms as targets for stress-induced depression (Supplementary Fig. 1d).

**In vitro screening for IL-6 inhibition and Rac1 promotion**. BDPP contains a variety of polyphenols. Orally consumed polyphenols are typically bioavailable in various organs and tissues as metabolites (phase II polyphenol conjugates and phenolic acids) following xenobiotic metabolism and gastrointestinal microbiome fermentation. It is possible that some metabolites may counteract the positive effect of the others. Therefore, if supplied as BDPP, the overall benefits might be reduced due to potential 'cancellation' effects. By identifying individual metabolites that selectively target key pathological mechanisms (e.g., Rac1 and IL-6) we can greatly improve the efficacy. We and others recently identified 21 BDPP-derived phenolic metabolites that accumulate in blood and/or in the CNS (14 polyphenolic metabolites and 7 phenolic acids, Supplementary Table 1)[25, 28, 33], among which 14 are currently accessible[28, 34]. We initiated a high-throughput screen of these metabolites for their ability to modulate IL-6 and Rac1. Peripheral blood mononuclear cells (PBMCs) isolated from C57BL/6 mice were pretreated with plasma bioavailable metabolites (Supplementary Table 1) for 16 h followed by stimulation with lipopolysaccharide (LPS) for 16 h. We found that 3(3,4-dihydroxy-phenyl) propionic acid (DHCA) was the most effective in reducing LPS-induced IL-6 production (Fig. 2a) with a calculated IC50 of 17.69 µM (Fig. 2b). In parallel studies, we screened 9 brain bioavailable metabolites using E18 MSN-enriched primary culture. We found that malvidin-3′-O-glucoside (Mal-gluc) significantly increased Rac1 expression (Fig. 2c) with a calculated EC50 of 3.52 nM (Fig. 2d). Among all the metabolites currently available for screening, we did not find any compounds that can simultaneously reduce IL-6 in PBMCs and increase Rac1 in primary MSN-enriched cultures.

**DHCA modulates intronic CpG methylation of IL-6**. To investigate how DHCA may attenuate IL-6 generation, we tested the effect of DHCA on Toll-like receptor-4 mediated signaling

pathways. PBMCs were pretreated with DHCA and stimulated with LPS using the same protocol as during screening. Samples were taken at 0, 30 min, 60 min, 6 h and 16 h following LPS stimulation and cell lysates were subjected to a multiplex ELISA assay. We found that DHCA had no effect on the activation of JNK or p38 as reflected by the lack of changes in phosphorylation of these molecules at any time points (Supplementary Figs. 2a and b). DHCA treatment produced a ~10% decrease in ERK and AKT activation 30 min and 60 min following LPS stimulation but no further changes were observed at 6 h and 16 h (Supplementary Figs. 2c and d). Based on evidence that DNA methylation at a single CpG in the human IL-6 promoter region effectively alters IL-6 expression[35], we next tested whether DHCA can modulate IL-6 gene expression through methylation mechanisms. We treated PBMC cells with DHCA and measured the expression of enzymes essential for DNA methylation/demethylation. We found DHCA treatment significantly reduced the expression of the DNA-methyltransferase 1 (DNMT1), an enzyme that plays a key role in methylation maintenance and also in de novo methylation processes (Fig. 2e). To investigate how methylation influences IL-6 gene expression, we employed the CpG-free luciferase reporter system. We cloned a ~280 bp CpG rich DNA segment from the mouse IL-6 promoter into a promoterless CpG-free reporter construct and transfected into N2a cells. We found the CpG-rich IL-6 promoter segment presented no inherent promoter activity as reflected by no differences in luciferase activity compared to the control construct (Fig. 2f). Moreover, treatment with 5-aza-2′-deoxycytidine (AZA-DC, a DNA methylation inhibitor) had no effect on the expression of luciferase (Fig. 2f), confirming methylation does not play any role in IL-6 promoter activity. We then cloned CpG rich DNA segments from IL-6 introns 1, 3 or 4 into the CpG-free luciferase reporter construct with a minimal EF1 promoter and no enhancer activity. Transfection of EF1-luciferase reporter constructs containing IL-6 intron 1 or intron 3 CpG-rich DNA segments significantly increased the luciferase activity, indicating both intronic 1 and 3 CpG-rich sequences present inherent enhancer activity (Fig. 2g). AZA-DC treatment partially reduced luciferase activity of intron 1 and totally abolished the enhancer activity of intron 3, demonstrating the contributions of intron 1 and intron 3 methylation on IL-6 transcription activity (Fig. 2g). Similar to AZA-DC, DHCA treatment also partially attenuated the enhancer activity of intron 1 and totally abolished the enhancer activity of intron 3 (Fig. 2g), suggesting DHCA has inhibitory activity on DNA methylation. In contrast, the CpG-rich region of IL-6 intron 4 presented no observable enhancer activity (Fig. 2g). These data suggest that DNA methylation at the CpG-rich sequences of IL-6 introns 1 and 3 can influence IL-6 expression and DHCA functions similar to a DNA methylation inhibitor that can reduce intronic DNA methylation to lead to attenuated IL-6 expression. To test whether DHCA may influence the expression of other pro-inflammatory factors, we measured the level of secreted cytokines following LPS stimulation (Supplementary Fig. 3). Besides IL-6, DHCA significantly reduced LPS-induced production of cytokines, mostly of pro-inflammatory nature, including granulocyte macrophage-colony stimulating factor (GM-CSF), IL-1β, IL-12 p40 and p70, IL-17 and MCP-1 (Supplementary Fig. 3) while in the absence LPS stimulation, DHCA did not influence cytokine expression profile.

**Mal-gluc increases acetylation of Rac1 gene promoter region**. Stress-induced reduction of Rac1 expression is associated with a repressed chromatin state surrounding the promoter region of Rac1 in the NAc[11]. Since Mal-gluc can cross the BBB, we hypothesized that Mal-gluc may promote Rac1 expression, in part, by modulating

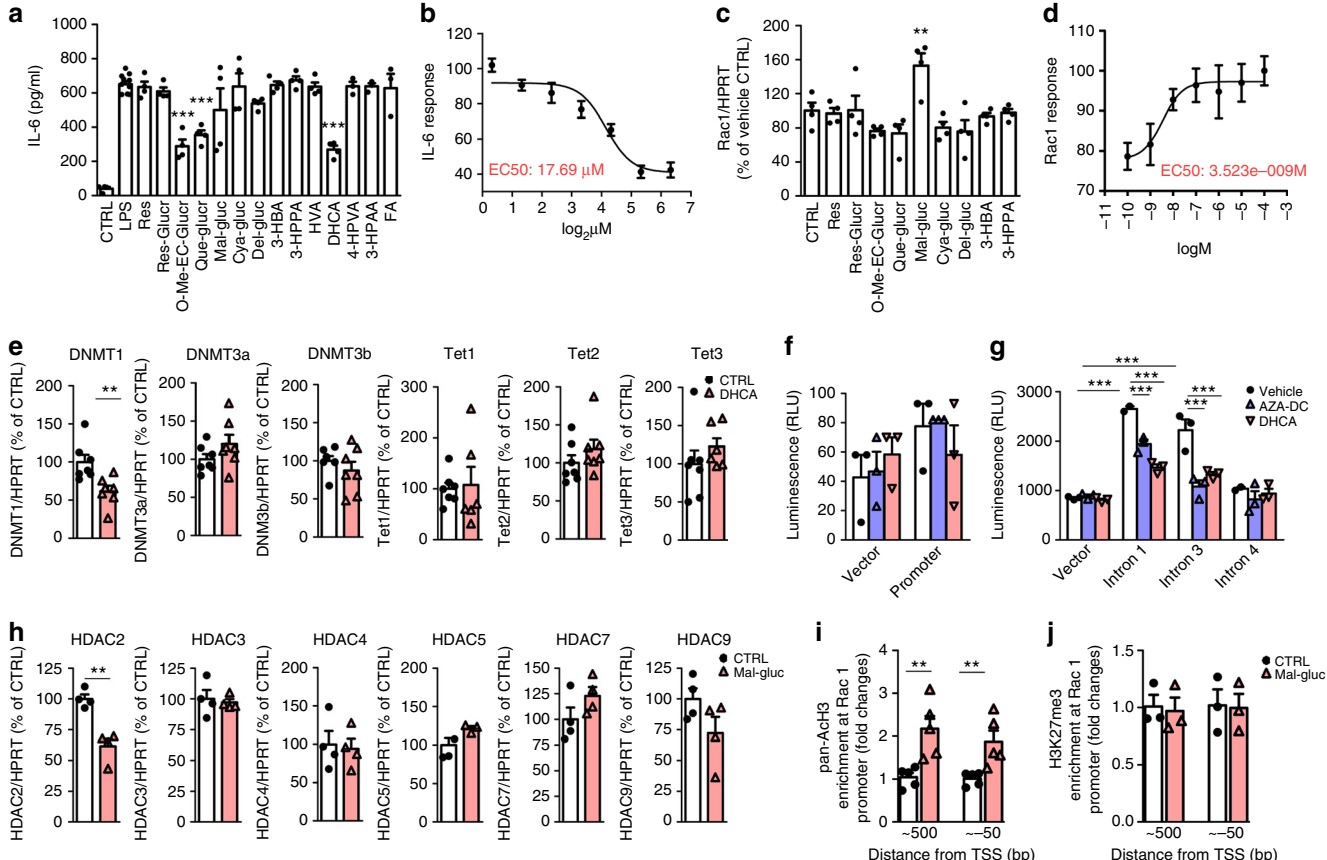

**Fig. 2** In vitro screening of phenolic metabolites to modulate IL-6 and Rac1 and mechanistic investigation of the role of DHCA on IL-6 and Mal-gluc on Rac1. **a**, **b** Screening of the effect of plasma bioavailable phenolic metabolites in inhibition of IL-6 in PBMCs following LPS stimulation. **a** Primary screening of 14 plasma bioavailable phenolic metabolites (one-way ANOVA, $F_{15,63} = 16.30$, $P < 0.0001$, $n = 3–8$ per culture condition). **b** Dose response and EC50 calculations of the effects of DHCA on IL-6. **c**, **d** Screening of the effects of brain bioavailable phenolic metabolites in promotion of *Rac1* expression in primary MSN-enriched cultures. **c** Primary screening of 9 brain bioavailable phenolic metabolites (one-way ANOVA, $F_{9,39} = 5.32$, $P = 0.0002$, $n = 4$ per culture condition). **d** Dose response and EC50 calculations of the effects of Mal-gluc on *Rac1* expression. **e** The expression of de novo methylation/demethylation genes in PBMCs following DHCA treatment (two-tailed unpaired t-test, $t_{12} = 3.220$, $P = 0.0074$ for *DNMT1*, $n = 7$ per culture condition). **f** Assessment of methylation-mediated promoter-like activity of CpG-rich sequences in the regions of IL-6 promoter in transfected N2A cells in the presence or absence of methylation inhibitor AZA-DC or DHCA (one-way ANOVA, $F_{5,17} = 1.278$, $P = 0.335$). **g** Assessment of methylation-mediated enhancer-like activities of CpG-rich sequences in the regions of IL-6 introns in transfected N2A cells in the presence or absence of methylation inhibitor AZA-DC or DHCA (one-way ANOVA, $F_{5,16} = 161.7$, $P < 0.0001$ for intron 1; $F_{5,17} = 25.54$, $P < 0.0001$ for intron 3; $F_{5,16} = 0.825$, $P = 0.558$ for intron 4). **h** HDAC genes expression in MSN-enriched primary culture following Mal-gluc treatment (two-tailed unpaired t-test, $t_6 = 2.781$, $P = 0.0017$ for *HDAC2*, $n = 4$ per culture condition). **i**, **j** Quantitative CHIP assessment of permissive H3 acetylation (**i**) and repressive trimethylation on H3K27 (**j**) along the mouse *Rac1* promoter and upstream in MSN-enriched primary neurons following Mal-gluc treatment (two-tailed unpaired t-test, $t_8 = 8.284$, $P = 0.0071$ for ~−500 bp upstream; $t_8 = 13.5$, $P = 0.0017$ for ~−50 bp in the promoter; $n = 5$ per culture condition for acetylation). All graphs represent mean ± s.e.m., **$P < 0.01$, ***$P < 0.001$

chromatin acetylation. We first assessed the effect of Mal-gluc on HDACs that play key roles in chromatin deacetylation. Treatment of MSN-enriched primary cultures with Mal-gluc significantly reduced the expression of HDAC2, but had no observable effect on other class I or class II HDACs (Fig. 2h). Using site directed quantitative chromatin immunoprecipitation (qCHIP), we examined the permissive histone H3 acetylation (AcH3) in Mal-gluc treated MSN-enriched primary cultures. Compared to vehicle, Mal-gluc significantly increased permissive acetylation of *Rac1* across the region surveyed (Fig. 2i) while having no effect on the repressive trimethylation of the *Rac1* promoter (Fig. 2j). Consistent with this observation, Mal-gluc treatment did not affect the expression of enzymes involved in de novo methylation and demethylation process (Supplementary Fig. 4).

**In vivo safety and dose finding**. We then initiated dose response and safety studies in mice to determine testing dosages and safety of DHCA and Mal-gluc. C57BL/6 mice were treated with either DHCA (doses ranging from 50 μg to 50 mg/kg-BW/day) or Mal-gluc (doses ranging from 50 ng to 5 mg/kg-BW/day) for 2 weeks to simulate long-term administration. Mice treated with DHCA were challenged with intraperitoneal (i.p.) injection of 0.4 mg/kg-BW LPS and plasma levels of IL-6 were measured 6 h post injection. Pretreatment with DHCA led to a dose-dependent suppression of IL-6 and that the groups treated with 5 and 50 mg/kg-BW/day showed the most significant reductions (Fig. 3a). In parallel, we measured *Rac1* mRNA in the NAc of animals treated with Mal-gluc by real-time PCR and found a dose-response increase of *Rac1* expression with 500 ng/kg-BW/day and 5 μg/kg-BW/day having the strongest promotion of *Rac1* (Fig. 3b). Following common discovery-stage practices, we measured the plasma levels of alkaline phosphatase (ALP), aspartate aminotransferase (AST), and alanine aminotransferase (ALT) for liver function and blood urea nitrogen (BUN) for renal function. We found neither phytochemical induced significant changes in any of the indexes (Supplementary Fig. 5).

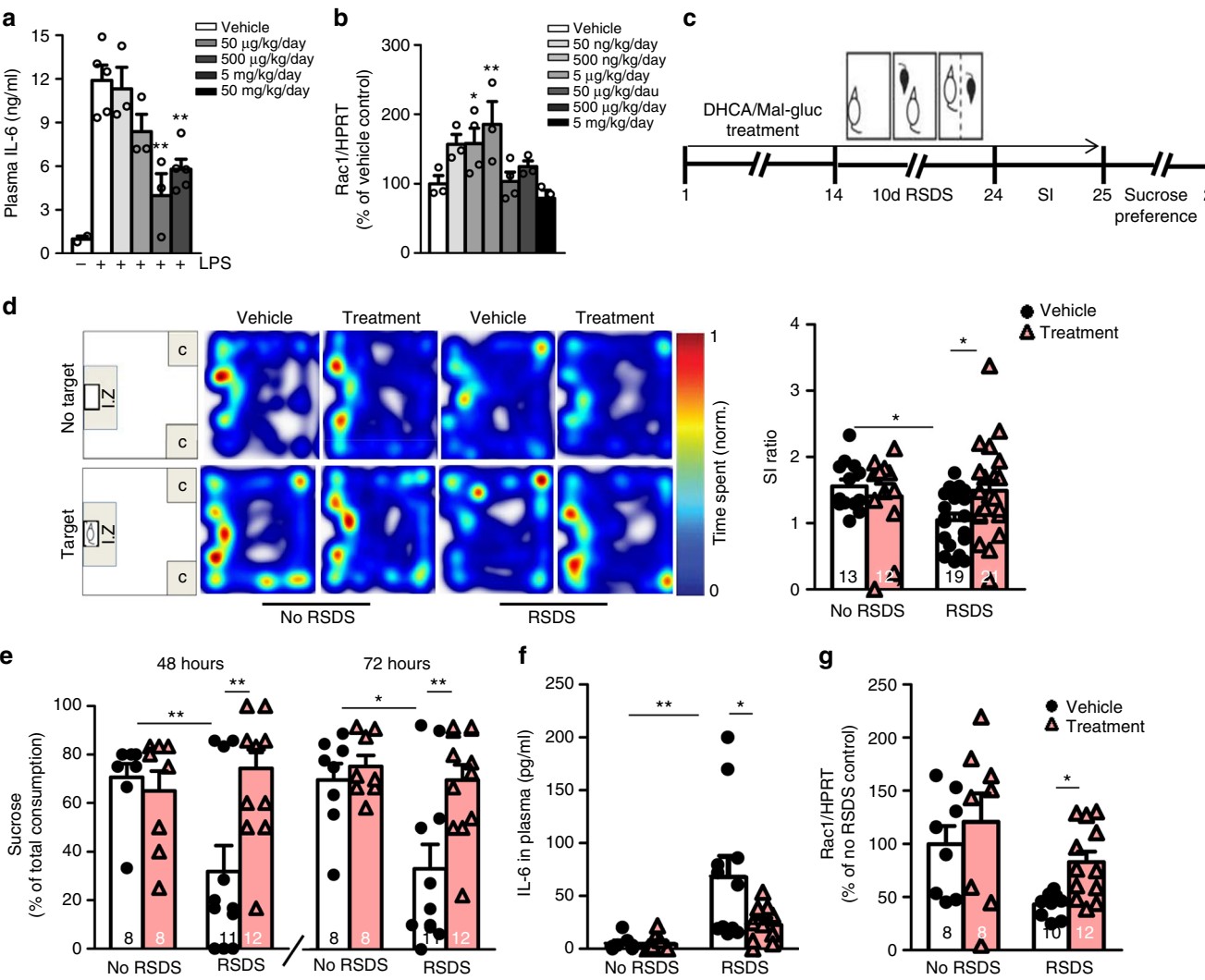

**Fig. 3** In vivo dose-finding and prophylactic effect of DHCA/Mal-gluc in promoting resilience to RSDS. **a** IL-6 levels in plasma 6 h post LPS challenge in mice treated with various doses of DHCA for 2 weeks (one-way ANOVA, $F_{5,20} = 12.50$, $P < 0.0001$, $n = 3$–5 animals per group). **b** Levels of *Rac1* mRNA in the NAc from mice treated with various doses of Mal-gluc for 2 weeks (one-way ANOVA, $F_{6,22} = 4.22$, $P = 0.01$, $n = 3$–4 animals per group). **c** Schematic design of the experiment. **d** Representative heatmaps and scatter plots of social avoidance behavioral test in mice treated with vehicle or DHCA/Mal-gluc for 2 weeks with or without RSDS (one-way ANOVA, $F_{3,64} = 2.79$, $P = 0.048$). **e** Sucrose preference test performed 48 h and 72 h following the SI testing (one-way ANOVA, $F_{3,38} = 5.53$, $P = 0.003$ for 48 h and $F_{3,38} = 6.73$, $P = 0.001$ for 72 h). **f** Plasma levels of IL-6 24 h following the last defeat (one-way ANOVA, $F_{3,35} = 6.25$, $P = 0.002$, $n = 7,8,11,10$ mice). **g** *Rac1* expression 48 h following the last defeat (one-way ANOVA, $F_{3,37} = 4.81$, $P = 0.007$). All graphs represent mean ± s.e.m., *$P < 0.05$, **$P < 0.01$

**Prophylactic treatment of DHCA/Mal-gluc promotes resilience**. Based on the dose–response studies, we chose 5 mg/kg-BW/day DHCA and 500 ng/kg-BW/day Mal-gluc for the preclinical studies. We tested the prophylactic effect by treating mice with a mixture of DHCA/Mal-gluc for 2 weeks prior to and throughout RSDS (Fig. 3c). We found that RSDS vehicle group had significantly lower SI ratios compared to unstressed mice while RSDS mice with DHCA/Mal-gluc treatment had significantly higher SI ratios (Fig. 3d). Following SI testing, mice were subjected to a sucrose preference test to evaluate the treatment effect on stress-induced anhedonia. We found that RSDS vehicle group had significantly reduced sucrose consumption compared to unstressed mice while DHCA/Mal-gluc treated RSDS mice had sucrose consumption similar to unstressed animals measured 48 and 72 h following the SI testing (Fig. 3e). These data suggest that DHCA/Mal-gluc treatment can prophylactically promote resilience against RSDS-induced anhedonia and social avoidance, both of which are key symptoms of depression in humans.

We next measured the plasma level of IL-6 following the defeat. RSDS led to a significantly higher induction of peripheral IL-6 and that treatment with DHCA/Mal-gluc significantly reduced plasma levels of IL-6 (Fig. 3f). Consistent with the in vitro studies (Fig. 2e), PBMCs isolated from stressed mice with DHCA/Mal-gluc treatment had significantly lower mRNA expression of both *IL-6* and *DNMT1* compared to vehicle-treated stressed mice (Supplementary Fig. 6). Examination of *Rac1* expression in the NAc revealed that RSDS led to reduced *Rac1* expression compared to the unstressed mice and DHCA/Mal-gluc treatment significantly prevented this decrease (Fig. 3g). These studies confirmed that treatment with DHCA/Mal-gluc led to the engagement of the intended targets.

To test whether DHCA/Mal-gluc-mediated modulation of IL-6 and Rac1 can lead to synaptic structural changes in the NAc, we measured the number of PSD-95 puncta in the NAc shell. Consistent with previous findings[14], we found that RSDS significantly increased the number of PSD-95 immunoreactive

puncta compared to the vehicle-treated unstressed mice (Fig. 4a). DHCA/Mal-gluc treatment significant reduced the number of puncta and the level was comparable to that from unstressed mice (Fig. 4a). To test the effect of DHCA/Mal-gluc on RSDS-induced synaptic functional changes, we measured the miniature excitatory postsynaptic currents (mEPSCs) in dopamine receptor D2 (Drd2) neurons in the NAc following RSDS using Drd2-EGFP transgenic mice. D2 neurons were selected because RSDS increases mEPSCs specifically on D2, but not D1 neurons in susceptible mice[32]. Drd2-EGFP mice were treated with DHCA/Mal-gluc for 2 weeks prior to and throughout the RSDS and subjected to SI testing. As expected, DHCA/Mal-gluc treatment significantly reduced the social avoidance phenotype (Fig. 4b). Electrophysiology recordings of mEPSCs of D2 neurons in the NAc shell showed that DHCA/Mal-gluc treatment significantly reduced mEPSC frequency compared to the vehicle treatment (Fig. 4c, d) with no difference in the mEPSC amplitude (Fig. 4e). Collectively, these data suggest that DHCA/Mal-gluc treatment may attenuate depression-like behavior in part through modulation of MSN synaptic structure and function in the NAc.

In parallel study, we tested the effect of single target compounds by treating mice with either 5 mg/kg-BW/day DHCA

or 500 ng/kg-BW/day Mal-gluc prior to RSDS. We found that neither treatment significantly promoted resilience (Supplementary Fig. 7a) or alleviate anhedonia (Supplementary Fig. 7b). These data indicate that combination treatment simultaneously targeting Rac1 and IL-6 is necessary for preventing stress susceptibility to RSDS.

**Therapeutic treatment of DHCA/Mal-gluc attenuates depression.** To test the therapeutic efficacy of DHCA/Mal-gluc, we induced depression-like behavior using two different methods. In the first set of studies, mice were subjected to RSDS followed by SI test. All susceptible mice with SI ratio <0.8 were randomly grouped into vehicle or DHCA/Mal-gluc group. Following two weeks treatment, all mice were subjected to SI re-testing without further stress exposure (Fig. 5a). We did not find statistical differences in the overall SI ratio (Fig. 5b), however, we found that 25% of the vehicle group displayed a resilient phenotype upon retesting whereas over 50% of the mice from the DHCA/Mal-guc group became resilient (Fig. 5b). In a parallel study to compare the treatment efficacy with the tricylic antidepressant Imipramine, we treated the susceptible mice with Imipramine through i. p. injection for 35 days. We found that about 50% of the

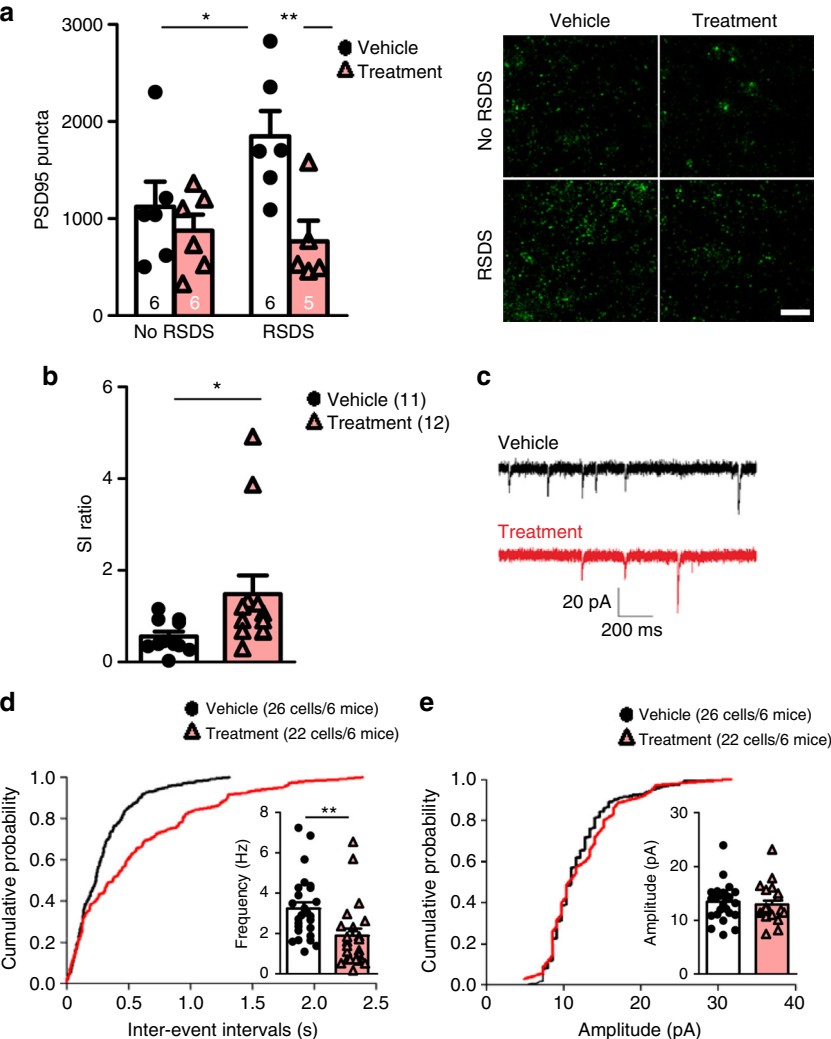

**Fig. 4** Prophylactic treatment with DHCA/Mal-gluc reverses RSDS-induced synaptic structural and functional alteration in the NAc. **a** Immunochemistry quantification of PSD95 puncta in NAc (one-way ANOVA, $F_{3,22} = 4.50$, $P = 0.015$). Inset; representative images of PSD95 puncta, scale bar = 10 μm. **b–e** DHCA/Mal-gluc treatment regulates synaptic transmission in D2 neurons in the shell of NAc following RSDS. **b** Social avoidance test in the Drd2 mice (two-tailed unpaired $t$-test, $t_{21} = 2.114$, $P = 0.047$). **c** Representative traces of mEPSCs. **d** mEPSCs frequencies (two-tailed unpaired $t$-test, $t_{46} = 2.890$, $P = 0.0059$). **e** mEPSCs amplitudes (two-tailed unpaired $t$-test, $t_{46} = 0.5008$, $P = 0.6189$). All graphs represent mean ± s.e.m., *$P < 0.05$, **$P < 0.01$

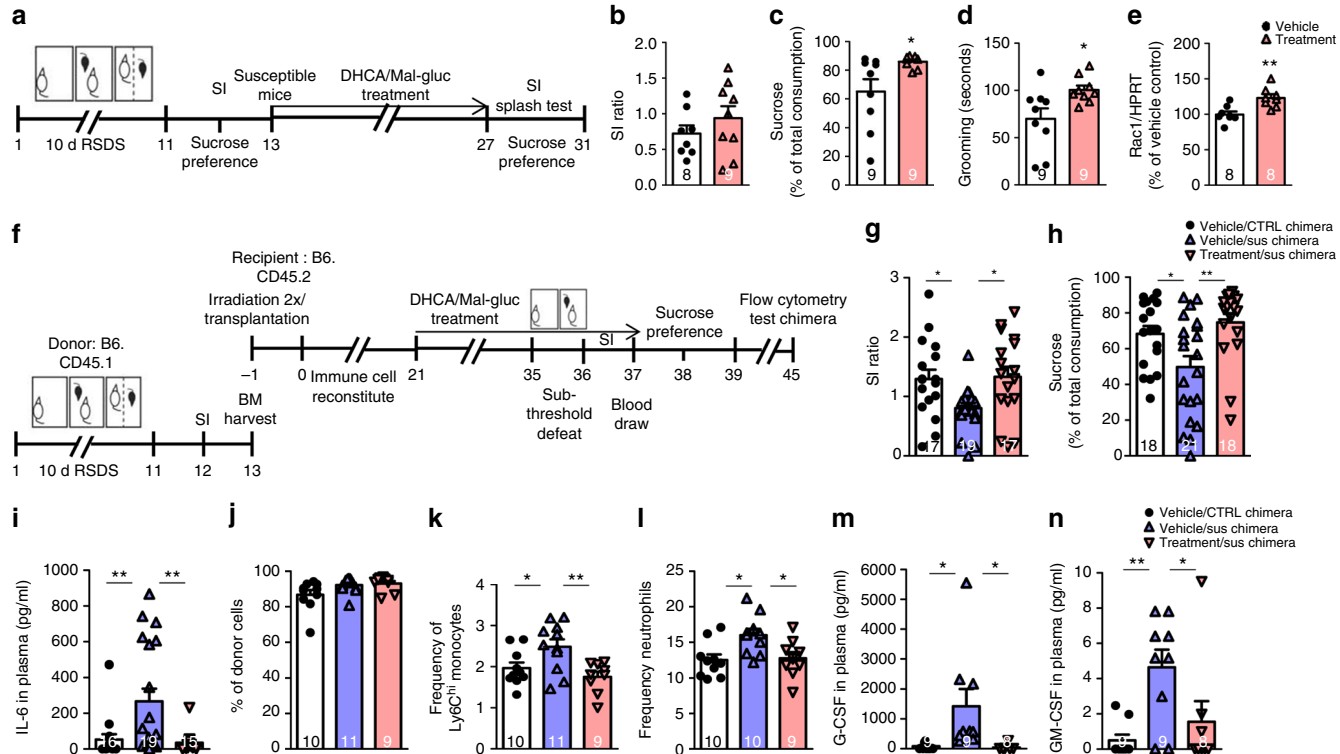

**Fig. 5** Therapeutic effect of DHCA/Mal-gluc in treating stress-induced depression. **a** Schematic design of the experiment. **b**–**d** Behavioral tests in stress susceptible mice following 2 weeks vehicle or DHCA/Mal-gluc treatment. **b** Social avoidance behavioral test (two-tailed unpaired t-test, $t_{15} = 1.039$, $P = 0.3152$). **c** Sucrose preference test (two-tailed unpaired t-test, $t_{16} = 2.406$, $P = 0.029$) and **d** splash test (two-tailed unpaired t-test, $t_{16} = 2.539$, $P = 0.021$). **e** Rac1 expression in the NAc (two-tailed unpaired t-test, $t_{14} = 3.637$, $P = 0.003$). **f** Schematic design of the experiment. **g**, **h** Behavioral tests following sub-threshold defeat in BM chimeras with BM reconstructed from naive mice or susceptible mice with or without DHCA/Mal-gluc treatment. **g** Social avoidance test (one-way ANOVA, $F_{2,52} = 4.58$, $P = 0.015$) and **h** sucrose preference test (one-way ANOVA, $F_{2,56} = 6.17$, $P = 0.004$). **i** Plasma level of IL-6 24 h after the sub-threshold defeat (one-way ANOVA, $F_{2,49} = 6.98$, $P = 0.002$). **j**–**l** DHCA/Mal-gluc treatment suppressed stress-induced increase of inflammatory Ly6C$^{hi}$ of monocytes and neutrophils in BM chimeras with BM reconstructed from susceptible mice. **j** Percentage of live leukocytes derived from the donor (one-way ANOVA, $F_{2,28} = 3.13$, $P = 0.144$). **k** Frequency of Ly6C$^{hi}$ monocytes of donor origin. Numbers represent percentages of live leukocytes (one-way ANOVA, $F_{2,29} = 6.18$, $P = 0.006$) and **l** frequency of neutrophils of donor origin. Numbers represent percentages of live leukocytes (one-way ANOVA, $F_{2,28} = 5.377$, $P = 0.011$). **m**, **n** Plasma levels of G-CSF and GM-CSF 24 h after the sub-threshold defeat (one-way ANOVA, $F_{2,25} = 5.189$, $P = 0.0138$ for G-CSF, $F_{2,25} = 6.103$, $P = 0.0075$ for GM-CSF). All graphs represent mean ± s.e.m., *$P < 0.05$, **$P < 0.01$

susceptible mice treated with Imipramine displayed a resilient phenotype upon retesting (Supplementary Fig. 8), suggesting that the efficacy of DHCA/Mal-gluc is comparable to that of Imipramine in treating RSDS-induced social avoidance phenotype. We then examined the effect of DHCA/Mal-gluc on RSDS-induced anhedonia and an ethologically relevant self-neglect phenotype. We found the average sucrose consumption was significantly higher in DHCA/Mal-gluc group compared to the vehicle group (Fig. 5c). Using the splash test, a measure of stress induced decreased self-care that is only reversible by chronic standard antidepressant treatment[36], we found that treatment with DHCA/Mal-gluc significantly increased the time spent grooming following aerosol delivery of a 10% sucrose solution to the fur (Fig. 5d). We next measured the levels of IL-6 in plasma and Rac1 in the NAc after behavioral testing. We found that circulating IL-6 was back to baseline levels two weeks after the defeat regardless of treatment status. This is largely consistent with our previous studies[30]. We found DHCA/Mal-gluc treatment significantly increased Rac1 expression in the NAc compared to the vehicle-treated group (Fig. 5e) suggesting that the therapeutic treatment indeed engages Rac1 in the NAc.

We also induced depression-like behavior in naive animals via a BM transplant from susceptible mice as a model of enhanced systemic inflammation[30]. As indicated in Fig. 5f, 4-week-old CD45.2$^+$ C57BL/6 recipient mice were irradiated and reconstituted with BM hematopoietic progenitors isolated from stress susceptible (average SI ratio of 0.46 from two mice) or unstressed control (average SI ratio of 1.81) donors. Following 3-week recovery, recipient mice were treated with vehicle or DHCA/Mal-gluc for 2 weeks before being subjected to a sub-threshold defeat stress that is not sufficient to induce social avoidance in control BM mice[11, 30]. As expected, vehicle-treated susceptible BM chimeras demonstrated increased social avoidance behavior compared to CTRL chimeras following sub-threshold defeat (Fig. 5g). Susceptible BM chimeras with DHCA/Mal-gluc treatment had a significantly decreased social avoidance behavior (Fig. 5g). The sucrose preference test showed that susceptible BM chimeras had significantly lower sucrose consumption compared to the CTRL chimeras and that this effect was completely reversed by DHCA/Mal-gluc treatment (Fig. 5h). Measurements of plasma IL-6 following sub-threshold defeat revealed that vehicle-treated susceptible BM chimeras had significantly higher levels of IL-6 compared to the CTRL BM chimeras[30], while treatment with DHCA/Mal-gluc significantly lowered IL-6 levels (Fig. 5i).

To determine the level of donor chimerism, blood was collected for flow cytometry analysis after the behavioral testing. We found that over 90% of the viable leukocytes were derived from the donor progenitor cells (Fig. 5j). Notably, DHCA/Mal-gluc treatment of susceptible BM chimeras significantly inhibited

stress-induced increases of donor-derived inflammatory Ly6C$^{hi}$ monocytes and neutrophils compared to vehicle-treated susceptible BM mice (Fig. 5k, l). Consistent with previous findings that cells from stressed or non-stressed donor do not affect BM cells reconstitution[37], we found no differences in the frequency of monocytes or neutrophils between the susceptible chimeras and the control chimeras prior to the defeat (Supplementary Figs. 9a and b) suggesting the increases of inflammatory monocytes and neutrophils were triggered by the sub-threshold defeat. We then measured the plasma levels of GM-CSF which stimulates the proliferation and release of granulocytes and monocytes from BM, and G-CSF, an important cytokine for the proliferation and differentiation of neutrophils, 24 h after the sub-threshold defeat. We found both G-CSF and GM-CSF were significantly higher in the vehicle-treated susceptible BM chimeras compared to the CTRL BM chimeras while DHCA/Mal-gluc treatment almost normalized the levels of both growth factors to that of the CTRL BM chimeras (Fig. 5m, n). These data suggest that DHCA/Mal-gluc can attenuate stress-induced upregulation of G-CSF and GM-CSF and subsequently the proliferation and release of monocytes and neutrophils from BM.

DHCA/Mal-gluc treatment itself did not have any effect on behavioral changes or circulating IL-6 in CTRL BM chimeras following sub-threshold defeat. Neither were there any changes in the number of inflammatory Ly6C$^{hi}$ monocytes or neutrophils (Supplementary Fig. 10).

**DHCA/Mal-gluc in variable stress (VS) model of depression.** RSDS is one of the best-established models for depression. However, no single model recapitulates all aspects of human depression. The utility and characterization of RSDS in females are also limited. To investigate whether DHCA/Mal-gluc can produce antidepressant responses to other stress paradigms, we used the VS model which has been shown to induce depression- and anxiety-like phenotypes in both male and female mice[36, 38]. Male mice were treated with DHCA/Mal-gluc for 14 days and subjected to a 21-day chronic variable stress (CVS) consisting of alternating foot shock, tail suspension and restrain (Fig. 6a). Following CVS, mice were subjected to a battery of behavior tests including the splash test, novelty suppressed feeding (NSF), forced swim test (FST), and sucrose preference test. As expected, CVS male mice groomed significantly less when sprayed with 10% sucrose solution compared to the non-CVS mice while DHCA/Mal-gluc treatment completely reversed CVS-induced self-neglect behavior (Fig. 6b). NSF test was used to examine an anxiety component of stress-induced behavior[39]. Stressed mice exhibited longer latency to feed compared to the non-stressed mice following overnight food deprivation. Notably, DHCA/Mal-gluc treatment significantly reduced the latency to eat (Fig. 6c). In a parallel control study, when food was provided in their home cage, there was no difference in feeding latency (Fig. 6c, inset). Next, FST was used to measure passive vs. active coping response. We found CVS mice spent significantly more time immobile compared to the non-stressed mice. Treatment with DHCA/Mal-gluc significantly reduced the time of floating (Fig. 6d). We found CVS did not induce anhedonia behavior using the sucrose preference test (Fig. 6e).

While 21 days of stress is necessary to induce depression-like behavior in male mice[38], female mice are more susceptible to variable stress and express a depression-associated phenotype

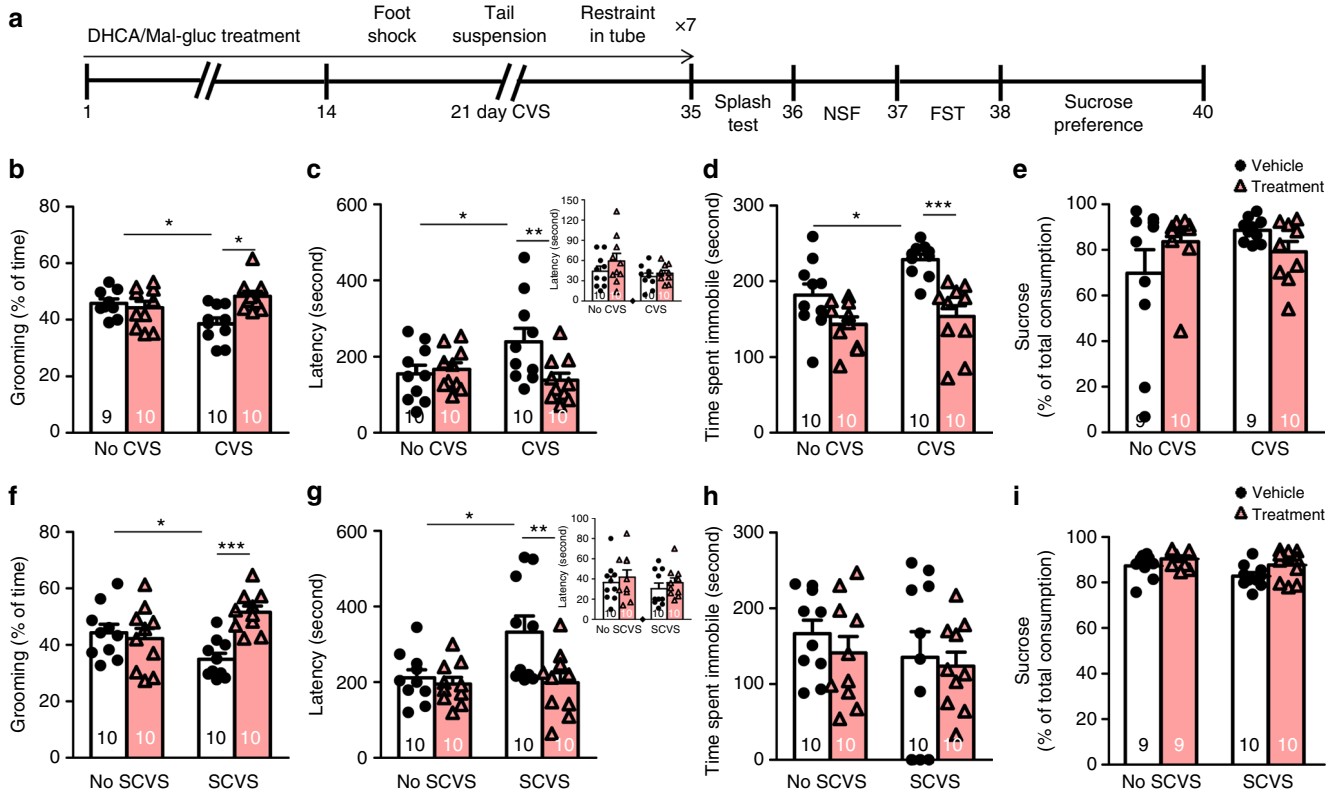

**Fig. 6** Prophylactic treatment with DHCA/Mal-gluc attenuates variable stress-mediated depression and anxiety phenotypes in both male and female mice. **a** Schematic design of the experiment. **b–e** Behavioral responses of male mice following 21 days of CVS. **b** Splash test (one-way ANOVA, $F_{3,38} = 4.61$, $P = 0.008$). **c** NSF test (one-way ANOVA, $F_{3,39} = 3.23$, $P = 0.034$). **d** FST (one-way ANOVA, $F_{3,39} = 10.30$, $P < 0.0001$) and **e** sucrose preference test (one-way ANOVA, $F_{3,37} = 1.65$, $P = 0.197$). **f–i** Behavioral responses of female mice following 6 days of SCVS. **f** Splash test (one-way ANOVA, $F_{3,39} = 5.33$, $P = 0.004$). **g** NSF test (one-way ANOVA, $F_{3,39} = 5.22$, $P = 0.004$); **h** FST (one-way ANOVA, $F_{3,39} = 0.581$, $P = 0.631$) and **i** sucrose preference test (one-way ANOVA, $F_{3,37} = 3.312$, $P = 0.032$). All graphs represent mean ± s.e.m., *$P < 0.05$, **$P < 0.01$, ***$P < 0.001$

after only 6 days of stress, termed subchronic variable stress (SCVS)[36, 40]. Similar to male mice, DHCA/Mal-gluc treatment significantly reduced depression-like behavior of self-neglect and anxiety in the splash and NSF tests (Fig. 6f, g). However, SCVS did not induce significant passive coping behavior or anhedonia in female mice (Fig. 6h, i).

**In vitro toxicity and drug-like properties.** Based on the efficacy of DHCA/Mal-gluc in treating stress-induced depression, we initiated an in vitro toxicity and drug-like properties studies to investigate the potential of developing these two phytochemicals as novel therapies. MTT assay and LDH assay showed that these two phytochemicals have no general cytotoxicity (Table 1). At low concentrations, DHCA and Mal-gluc had no detectable effects on cell number, nuclear size, DNA structure, cell membrane permeability, mitochondrial mass, mitochondrial membrane potential or cytochrome C release in high content screening (HCS)[41] using HepG2 cells (Table 1). At high concentrations, Mal-gluc reduced cell number, increased nuclear size, reduced cell membrane permeability, and increased mitochondrial mass, with calculated EC50s beyond physiological concentration (Table 1). DHCA, at high concentrations also decreased cell membrane permeability, increased mitochondrial mass and cytochrome C release (Table 1). IC50 of both phytochemicals for inhibition of the human Ether-à-go-go-related gene (hERG)[42] was >25 μM (Table 1). Based on common discovery-stage cutoff criteria that HCS cytotoxicity with EC50 > 50 μM and inhibition of hERG with IC50 > 20 μM are acceptable, both phytochemicals are deemed to possess acceptable cytotoxicity/cardiotoxicity properties.

For drug-like properties, we assessed the two phytochemicals for plasma stability[43], brain stability[44], plasma protein binding[45] and inhibition of cytochrome P450s (CYPs) (Table 1). Common discovery-stage cutoff criteria used for classifying individual drug-like properties as acceptable are: T1/2 > 30 min for plasma stability, T1/2 > 30 min for brain stability, plasma protein binding >2% free (i.e., <98% bound fraction) and IC50 > 10 μM for any CYP isoform. Thus, both phytochemicals are deemed to have acceptable drug-like properties.

**Monoaminergic receptor and transporter binding activities.** Currently available antidepressants are believed to act by modifying the activity of the brain monoaminergic system[46, 47]. We evaluated whether Mal-gluc and DHCA directly interact with the monoaminergic system using a radioligand binding assay. Among the 37 receptors and transporters tested (Table 2), we found that DHCA has no significant binding with any of them at concentrations as high as 10 μM (Table 2). Mal-gluc, at 10 μM concentration, binds weakly to 5-HT1D and 5-HT3 with a calculated inhibitory constant (Ki) of 4.3 μM for 5-HT3 and >10 μM for 5-HT1D, suggesting Mal-gluc does not have significant effect on the serotonin receptors at physiological concentrations. Collectively, these data suggest that neither phytochemical directly interacts with receptors/transporters in the brain that are known to play key roles in the pathogenesis of depression.

## Discussion

Currently available antidepressant treatments are mainly designed to target the serotonergic and/or noradrenergic system in the brain. Given the relatively low overall response rates and the wide range of 'adverse' events associated with these treatments, there is an urgent need for new therapeutics to treat specific underlying disease mechanisms that are not addressed by standard antidepressants and simultaneously treat multiple pathogenic mechanisms to increase the likelihood of therapeutic efficacy.

**Table 1 In vitro toxicity characteristics and drug-like properties of DHCA and Mal-gluc**

| | Toxicity and drug-like property assays | Malvidin-gluc | DHCA |
|---|---|---|---|
| In vitro assays for toxicity | *MTT assay* | | |
| | % of vehicle control | 100% | 100% |
| | *LDH assay* | | |
| | % of vehicle control | 100% | 100% |
| | *Cytotoxicity by high content screening (AC50)* | | |
| | Cell number | 176 μM ↓ | NR |
| | Nuclear size | 88.4 μM ↑ | NR |
| | DNA structure | NR | NR |
| | Cell membrane permeability | 52.7 μM ↓ | 56.6 μM ↓ |
| | Mitochondrial mass | 176 μM ↑ | NR |
| | Mitochondrial membrane potential | NR | >200 μM ↑ (NS) |
| | Cytochrome C release | NR | >200 μM ↑ |
| | *Cardiotoxicity (IC50)* | | |
| | hERG inhibition | >25 μM | >25 μM |
| In vitro assays for drug-like properties | *CPY450 inhibition (IC50)* | | |
| | CYP2D6 | >25 μM | >25 μM |
| | CYP3A4 | >25 μM | >25 μM |
| | CYP1A2 | >25 μM | >25 μM |
| | CYP2C9 | >25 μM | >25 μM |
| | CYP2C19 | >25 μM | >25 μM |
| | CYP2C8 | >25 μM | >25 μM |
| | CYP2B6 | >25 μM | >25 μM |
| | *Plasma stability* | | |
| | Half-life (mouse plasma) | 201 min | 44.1 min |
| | Half-life (human plasma) | 250 min | 66.7 min |
| | *Brain homogenate stability* | | |
| | Half-life (mouse homogenate) | 29.4 min | 16.1 min |
| | *Plasma protein binding* | | |
| | Mean fraction unbound (mouse plasma) | 55.70% | N/A |

NR, No response observed; N/A, Not performed due to the short half-life in plasma; AC50, The concentration at which 50% maximum effect is observed for each cell health parameter; IC50, The concentration at which 50% maximum inhibition is observed for each parameter; ↓↑, Decrease or increase of response; NS, Fit not statistically significant

Natural products have a history of being the source for many of the active ingredients in medications[48, 49]. In recent years, phytochemicals have received growing interest due to their strong antioxidant, anti-inflammatory, antimicrobial, and antitumorigenic activities. Previously, we have shown that BDPP is beneficial in various animal models of cognitive dysfunction. However, the effect of BDPP on depression was never before tested. Here we demonstrated that oral administration of BDPP is effective in attenuating the development of depression-like behaviors in a well-established RSDS model in mice. Moreover, we identified Mal-gluc and DHCA, two bioavailable metabolites derived from xenobiotic metabolism and gut microbiome metabolism of BDPP, can prophylactically prevent as well as therapeutically treat RSDS-induced depression phenotypes. Our evidence suggests that both DHCA and Mal-gluc contribute to resilience against the development of depression-like phenotypes by modulating, respectively, pathological mechanisms relating to IL-6 and Rac1. In particular, Mal-gluc significantly promotes Rac1 expression by reducing HDAC2 expression and, thereby, increasing histone acetylation along Rac 1 promoter and upstream gene sequences while DHCA significantly inhibits PBMC IL-6 expression by inhibiting DNA methylation of the IL-

**Table 2 Binding activities of DHCA and Mal-gluc with monoaminergic receptors and transporters**

| Receptor | 5-HT1A | 5-HT1B | 5-HT1D | 5-hT1e | 5-HT2A | 5-HT2B | 5-HT2C | 5-HT3 | 5-hT5a | 5-HT6 | 5-HT7 |
|---|---|---|---|---|---|---|---|---|---|---|---|
| DHCA | 3.3 | −11.8 | 11.7 | −1.7 | 18 | −7 | −9.9 | −11.7 | 4.2 | 24.2 | 4.2 |
| Mal-gluc | 8.6 | 10.4 | 54 | 10.3 | −10.5 | −13.8 | −6.9 | 74.1 | 5.2 | −18.5 | −13.5 |
| **Receptor** | **Alpha1A** | **Alpha1B** | **Alpha1D** | **Alpha2A** | **Alpha2B** | **Alpha2C** | **Beta1** | **Beta2** | **Beta3** | | |
| DHCA | 4 | 11.1 | −15.4 | 15.7 | −2.3 | 27.3 | −14.1 | −7.6 | −15.8 | | |
| Mal-gluc | −4.5 | 24.4 | −1.3 | 7.3 | −5.9 | −11 | −15.5 | 1.9 | −17.7 | | |
| **Receptor** | **M1** | **M2** | **M3** | **M4** | **M5** | **BZP RBS** | **PBR** | **DOR** | **MOR** | | |
| DHCA | −1.8 | −12.2 | 9.4 | 3 | 21.2 | −5.2 | −13.3 | 9.5 | −7.1 | | |
| Mal-gluc | −6.2 | 5.2 | 23.6 | −4.6 | −10.2 | 5.4 | 15.7 | −10.2 | −5 | | |
| **Transporter** | **D1** | **D2** | **D3** | **D4** | **D5** | **DAT** | **NET** | **SERT** | | | |
| DHCA | −3.7 | −5.7 | −18.8 | 1.9 | 5.1 | 34.5 | −16.4 | −0.9 | | | |
| Mal-gluc | −10.3 | 7.2 | 7.2 | −2 | −2 | −18.2 | −9.4 | 7.8 | | | |

Crude membrane fractions prepared from a stable human embryonic kidney 293 (HEK) cell line expressing recombinant receptors were mixed with radio-labeled specific ligand and 10 μM of competitor compound (DHCA, Mal-gluc, or non-specific ligand). Percentage radioligand binding was measured. Compounds that show a minimum 50% inhibition were subjected to a secondary radioligand binding assay to determine equilibrium binding. Data points represent means from quadruplicate reactions. Alpha, alpha adrenergic receptor; Beta, beta adrenergic receptor; BZP RBS, rat brain benzylpiperazine site; D, dopamine receptor; DAT, dopamine transporter; DOR, δ-opioid receptor; 5-HT, serotonin receptor; M, muscarinic receptor; MOR, μ-opioid receptor; NET, norepinephrine transporter; PBR, peripheral benzodiazepine receptor; SERT, serotonin transporter

6 genes at the CpG-rich sequences of IL-6 introns 1 and 3. Notably, besides IL-6, DHCA is also capable of modulating other inflammatory cytokines including IL-1β and IL-12, both of which have been reported to be elevated in MDD subjects. Whether these changes are the direct effect of DHCA or secondary to IL-6 induction, and their potential contribution to attenuate stress-induced depression need further investigation. Epigenetic modulation of chromatin structure and subsequent gene expression alteration following chronic stress both contribute to the pathogenesis of stress-related disorders and depression. Previous evidence demonstrated that RSDS induces a transient decrease in cellular levels of histone H3 acetylation in the NAc and pharmacological inhibition of HDACs or genetic manipulation of HDAC2 leads to global normalization of stress-induced aberrant gene expression and exerts potent anti-depressant-like effects in behavior testing[50, 51]. However, the development of currently available HDAC inhibitors for treating psychiatric disorders is largely hindered by their lack of specificity and limited BBB penetration. In comparison, Mal-gluc can be a promising agent for treating depression since it is capable of penetrating across the BBB and accumulating in the brain, and that Mal-gluc selectively inhibits HDAC2 while sparing other classes of HDACs.

Depression is a multifaceted disease with many underlying mechanisms. We rationalized that our approach to use a combination of DHCA and Mal-gluc to simultaneously inhibit peripheral inflammation and modulate synaptic plasticity in the NAc would work synergistically to optimize resilience against chronic stress-induced depression-like phenotypes (Supplementary Fig. 1d). Our observation is consistent with clinical and preclinical evidence that overly active peripheral inflammation processes involving inflammatory cytokines such as IL-6 and disruptions in the normal synaptic plasticity responses in the NAc are two key pathological mechanisms underlying depression and anxiety. This is further supported by our evidence that single compound treatment is not sufficient to modulate RSDS-induced depression-like phenotypes. While the mechanisms underlying the interaction between peripheral pro-inflammatory cytokines and depression are largely unknown, we observed here that selective removal of IL-6, specifically from bone marrow-derived leukocytes, significantly blocked the stress-induced modulation of excitatory synapses in the NAc. Our evidence demonstrated a cause-effect relationship among leukocytes-derived proinflammatory responses, brain reward circuitry synaptic remodeling, and the manifestation of depression-like behavioral phenotypes, which supports the consideration of IL-6 and IL-6 producing cells (and perhaps additional pro-inflammatory

molecules), as a key therapeutic target for treating depression. This is further supported by recent findings that social defeat stress induces neurovascular pathology and increases BBB permeability which may facilitate larger molecules such as IL-6 and possibly immune cells infiltration into the CNS[52]. The effect of DHCA/Mal-gluc on stress-mediated inflammation is also confirmed in the mouse model of increased systemic inflammation that is induced by transplantation of hematopoietic progenitor cells from stress-susceptible mice. Our data demonstrated that treatment significantly attenuated depression-like phenotypes and reduced peripheral IL-6 following sub-threshold defeat in the chimeric mice and this was accompanied by decreased number of circulating inflammatory monocytes and neutrophils. Previous studies showed that both in humans and in rodents, chronic stress induced monocytosis and neutrophilia are mediated, in part, by increased activation of the sympathetic nervous system involving the β3 adrenergic receptor[37]. However, we found that neither DHCA nor Mal-gluc can effectively interact with β3 or other adrenergic receptors. In contrast, we observed that DHCA/Mal-gluc treatment significantly reduced the levels of circulating GM-CSF and G-CSF, both are important cytokines for the proliferation and migration of granulocytes and monocytes.

Sex differences in depression are well documented both in humans and in animal models. Current depression diagnosis and epidemiology studies showed that women are twice as likely to develop depression as men[53]. However, the majority of the preclinical studies in developing therapeutics have primarily used male mice[54]. In this study, we tested the efficacy of DHAC/Mal-gluc in both sexes using another well-established VS mouse model of depression. We found that DHAC/Mal-gluc is effective in alleviating non-social stress induced depression-like behavior including self-neglect and anxiety in both male and female mice, further confirming the efficacy of the treatment. We found that 21-day VS induced a passive coping response in male mice that was reversed by the treatment while female mice exposed to a shorter 6-day VS did not exhibit the same passive coping response. This is consistent with experimental observation that males and females may manifest different symptoms and responses toward stress and further emphasizes the importance of testing any therapeutics in both sexes.

The RSDS paradigm we used in this study leads to long-lasting behavioral consequences in mice that recapitulate many key behavioral features that are associated with psychosocial stress in humans[55]. Similar to human psychopathology, chronic social subordination of susceptible mice leads to a spectrum of depression-like behaviors, among which social avoidance and

anhedonia are most relevant to human depression. The efficacy of DHCA/Mal-gluc in alleviating these depression-like symptoms may also be suitable for treating other neuropsychological disorders such as posttraumatic stress disorder, traumatic brain injury-induced mood disorder and bipolar depression, which share similar symptoms with MDD. We demonstrated that through mechanisms involving epigenetic modification, DHCA/Mal-gluc can effectively target both inflammation and brain synaptic plasticity which are not addressed by the classical antidepressants[1, 47]. Our evidence supports the development of DHCA/Mal-gluc as a novel therapeutic agent to treat patients with treatment-resistant MDD, particularly among the majority of patients who are characterized by high plasma levels of IL-6[30]. Moreover, because neither Mal-gluc nor DHCA effectively interacts with monoaminergic systems that are targeted by classical antidepressants, DHCA/Mal-gluc can be safely used in combination with currently available antidepressants to simultaneously target multiple disease mechanisms and increase the likelihood of therapeutic success. Given the safety and drug-like profile, and the lack of direct interaction with key molecular components of the monoaminergic system, and the demonstrated efficacy in both males and females in experimental model, DHCA/Mal-gluc can immediately translate into human clinical studies for the treatment of stress disorders and depression either alone or in combination with currently available antidepressants.

## Methods

**Materials**. Resveratrol (ChromaDex, Irvine, CA, USA), GSPE (Warehouse, UPC: 603573579173), CGJ (Welch), Mal-gluc, cyanidin-3-*O*-glucoside, delphinidin-3-*O*-glucoside, quercetin-3′-*O*-glucuronide and resveratrol-3′-*O*-glucuronide (Extrasynthesis, Genay Cedex, France), 3-hydroxybenzoic acid, 3-(3′-hydroxyphenyl) propionic acid, homovanilic acid, DHCA, 5-(4′-dydroxyphenyl) valeric acid, 3-hydorxyphenylacetic acid, ferulic acid-4′-*O*-sulfate (Sigma-Aldrich) were obtained commercially and 3′-*O*-methyl-epicatechin-5′-*O*-glucuronide was synthesized as previously described[28, 34]. All tested compounds were analyzed by LC-MS and archived as previously reported[26, 33] in compliance with NCCIH Product Integrity guidelines.

**Animals**. Male CD45.2+ C57BL/6 mice were used for RSDS and as bone marrow (BM) recipients. CD45.1+ C57BL/6 mice were used as BM donors, except for in the IL-6−/− transplant study, which by necessity hosts were CD45.1+ C57BL/6 mice. All C57BL/6 mice were purchased from the Jackson Laboratory. Retired breeder CD-1 mice were purchased from Charles River Laboratory. Drd2-EGFP mice were bred and genotyped[56]. All animals had access to regular chow ad libitium and were maintained on a 12:12 light/dark cycle with lights on at 07:00 h in a temperature-controlled (20 ± 2 °C) vivarium and all procedures were approved by the Institutional Animal Care and Use Committee.

**BDPP treatment**. Male C57BL/6 mice (8-week-old) were randomly grouped into two groups: one group received regular drinking water; the other group was treated with BDPP composed of GSPE, RESV and CGJ, delivered through their drinking water, starting 2 weeks prior to RSDS and throughout RSDS and SI testing. The calculated daily intake of GSPE was 200 mg/kg body weight (BW), RESV was 300 mg/kg BW and CGJ was 1 ml/day.

**DHCA/Mal-gluc treatment**. For prophylactic treatment, 8-week-old male C57BL/6 mice were randomly grouped into two groups: one group was treated with vehicle, the other group was treated with a mixture of Mal-gluc (0.5 μg/kg-BW/day) and DHCA (5 mg/kg-BW/day), delivered through their drinking water, starting 2 weeks prior to the RSDS and throughout RSDS. To facilitate the follow-up sucrose preference test, the delivery of the drinking solution was through two 50-ml tubes with sipper top for each cage, starting from the day of initiation of treatment and the two tubes were regularly switched during the treatment. In parallel control studies, mice received similar treatment but were not subjected to RSDS. For therapeutic studies, susceptible mice (SI <0.8) were identified and randomly grouped into two groups either treated with vehicle or DHCA/Mal-gluc for 2 weeks. Mice were subjected to SI, sucrose preference, and splash testing. For testing the therapeutic efficacy of Imipramine, susceptible mice were treated with Imipramine (20 mg/kg-BW/day, i.p.) for 34 days. On day 35, mice were tested for SI 15–20 min after the last injection.

**Generation of BM chimeras and treatment**. Generation of BM chimeras was performed as previously described[57]. Briefly, 4-week-old recipient CD45.2+ C57BL/ 6 mice were lethally irradiated with 1200 rad delivered in two doses of 600 rad each (10–11 h apart). Following the second irradiation, BM hematopoietic cells from the donor mice were introduced through a retro-orbital injection. BM cells were obtained from the naive or susceptible CD45.1+ C57BL/6 mice. All mice were treated with antibiotics (sulfatrim) for three weeks followed by two-week treatment with vehicle or DHCA/Mal-gluc before being subjected to a sub-threshhold defeat.

**In vivo safety, toxicology analyses and dose finding studies**. Male C57BL/6 J mice (8-week-old) were placed on a polyphenol-free AIN-93M diet and treated with various doses of DHCA (vehicle, 50 μg, 500 μg, 5 mg, and 50 mg/kg-BW/day), delivered through their drinking water for two weeks. Animals were then sacrificed and plasma was collected for routine biochemistry panel analysis including ALP, ALT, and AST for liver function and blood urea nitrogen (BUN) content for kidney function. A separate group of mice was similarly treated and challenged with an i.p. injection of 0.4 mg/kg body weight LPS and plasma was collected 6 h post injection for IL-6 analysis. For Mal-gluc study, mice were treated with vehicle, 50 ng, 500 ng, 5 μg, 50 μg, 500 μg and 5 mg/kg-BW/day, delivered through their drinking water for 2 weeks. Mice were sacrificed and plasma was collected for ALP, ALT, AST and BUN analysis and NAc was isolated for gene expression analysis. For both groups of mice, tissues were collected for pathology evaluation.

**RSDS**. CD-1 mice were screened for aggressive prior to the start of social defeat experiments based on previously described criteria[55], and housed in one side the social defeat cage (26.7w × 48.3d × 15.2h cm; Allentown Inc) divided by a clear, perforated Plexiglass divider 24 h prior to the start of defeats. Mice subjected to RSDS were exposed to a novel CD-1 aggressor mouse for 10 min once per day, over 10 consecutive days. Following the 10 min of interaction, the experimental mice were moved to the opposite side of the defeat cage and kept sensory contact with CD-1 mice through the perforated divider until the next defeat. Control mice without RSDS were housed two mice per cage, on opposite sides of the perforated divider, rotated daily in a manner similar to the defeat group, but never exposed to aggressive CD-1 mice. Mice were returned to a single house following the last defeat[55, 58].

**Subthreshold defeat stress**. C57BL/6 mice were subjected to a novel CD-1 aggressor for three consecutive 5-minutes defeat bouts, with a 15 min inter trial interval between the defeats. Mice were subjected to social avoidance testing 24 h after the last interaction[11, 55, 58]. Under control conditions, this protocol does not result in social avoidance behavior.

**Variable stress**. VS was performed as described[36, 40]. Specifically, female mice underwent 1 h of variable stress each day for 6 days while male mice underwent VS for 21 days. VS consists of foot shock, tail suspension, and restraint. To prevent habituation, stressors were administered in the following order: 100 random mild foot shocks lasting 2 s at 0.45 mA for 1 h (Med Associates, St. Albans, Vermont, USA); tail suspension stress for 1 h; restraint stress—mice were placed inside a 50-ml falcon tube for 1 h within the home cage. The three stressors were then repeated 1 × for the duration of three days (for female mice) and 6 × for the duration of 18 days (for male mice) in the same order. After each stress, animals were returned to their home cage.

**Social avoidance test (Social interaction test)**. All social interaction (SI) tests were performed under red-light conditions. Mice were placed in a novel interaction open-field arena (42 × 42 × 42 cm; Nationwide Plastics) with a small animal cage placed at one end. Their movements were automatically monitored and recorded (Ethovision 3.0; Noldus Information Technology) for 2.5 min in the absence (target absent phase) of a novel CD-1 mouse and followed by 2.5 min exploratory behavior in the presence of a caged CD-1 mouse (target present phase). SI behavior was then calculated as a ratio of the time spent in the interaction zone with the target present divided by the time spent in the interaction zone with the target absent. All mice with a ratio above 1.0 were classified as resilient whereas below 1.0 were classified as susceptible[55].

**Splash test**. Splash test was carried out in a standard mouse cage with no bedding under a red light[59]. Briefly, mice were sprayed with 200 μl of a 10% (wt/vol) sucrose solution directly onto the animal's back using a small atomizer to induce grooming behavior. The grooming frequency and latency were recorded for 5 min and manually scored.

**Novelty suppressed feeding (NSF)**. Mice were food restricted overnight before testing. On the day of testing, mice habituated to the testing room for 1 h. Under red light conditions, mice were then placed into a plastic box 50 × 50 × 50 cm with bedding. A single pellet of food was placed in the center of the box. Mice were placed in the corner of the box, and the latency to eat was scored up to 10 min during testing. Mice were then immediately transferred to their home cage in standard lighting conditions, and the latency to eat was recorded.

**Forced swim test (FST).** FST was conducted as previously described[36, 40]. Twenty-four hours after the NSF test, animals were acclimated in the test room for an hour. Mice were then tested in a 4 liter Pyrex glass beaker, containing 2 liters of water at 25 + 1 °C for 6 min. Behavior was videotaped and hand scored for percentage time spent immobile by an observer blind to experimental conditions.

**Sucrose preference testing.** Mice were given access to a two-bottle choice of water or 1% sucrose solution and the consumption of each solution was recorded once every 24 h for 48 or 72 h. The two bottles were switched following each recording. Sucrose preference was calculated as a percentage of sucrose consumption over total liquid consumption.

**Electrophysiology.** Male Drd2-GFP mice were perfused with ice-cold artificial cerebrospinal fluid (aCSF) containing (mM): 128 mm NaCl, 3 KCl, 1.25 NaH$_2$PO$_4$, 10 d-glucose, 24 NaHCO$_3$, 2 CaCl$_2$, and 2 MgSO$_4$, pH 7.35 (oxygenated with 95% O$_2$ and 5% CO$_2$, 295–305 mOsm). Acute brain slices containing the NAc were cut using a microslicer (DTK-1000, Ted Pella) in sucrose-aCSF, which was derived by fully replacing NaCl replaced by 254 sucrose[60, 61]. Electrophysiological recordings were performed at 30–32 °C in aCSF containing 50 μm picrotoxin to block GABA$_A$ receptor-mediated IPSCs and 1.5 μm tetrodotoxin to block action potentials. Patch pipettes (3–5 MΩ resistance) were filled with an internal solution containing the following (mM): 115 potassium gluconate, 20 KCl, 1.5 MgCl$_2$, 10 HEPES, 10 phosphocreatine, 2 ATP-Mg, 0.5 GTP, and 1 QX-314 [2(triethylamino)-N-(2,6-dimethylphenyl) acetamine, a Na$^+$ channel blocker], pH 7.2 (280–290 mOsm). The shell of the NAc was identified under visual guidance using infrared differential interference contrast microscopy (Olympus BX51-WI). Whole-cell voltage-clamp recordings at a holding potential of −80 mV were performed in GFP$^+$ cells with a computer-controlled amplifier (MultiClamp 700B, Molecular Devices), digitized (Digidata 1440 A, Molecular Devices), and acquired (pClamp 10.3, Molecular Devices) at a sampling rate of 10 kHz. Only cells with resting membrane potential of −71 to −81 mV were included for analysis. The frequency and amplitude of mEPSCs were analyzed using Minianalysis software (Synaptosoft).

**Blood sample collection.** Submandibular vein bleeds[62] were taken from mice 24 h following the SI test and plasma was collected by centrifugation at 3000 r.p.m. for 15 min. Blood from chimeras was collected into ice-cold FACS buffer (Ca$^{2+}$ Mg$^{2+}$-free PBS supplemented with 2% heat inactivated FBS and 5 mM EDTA), red blood cells were lysed using RBC lysis buffer (Biolegend) and cells were stained with antibodies for flow cytometry assessment.

**Flow cytometry and cell type analysis.** Flow cytometry studies were performed using a LSRII Fortessa (Becton Dickinson) and analyzed using FlowJo software (Tree Star). Fluorochrome or biotin-conjugated mAbs specific for mouse B220 (clone RA3-6B2), CD11b (clone M1/70), CD45.1 (clone A20), CD45.2 (clone 104), CSF-1R (also called CD115) (clone AFS98), Ly6C (clone HK1.4), Ly6G (clone 1A8), CD3 (clone 17A2), and the secondary reagents (allophycocyanin, peridinine chlorophyll protein, and phycoerythrin-indotricarbocyanine-conjugated streptavidin) were obtained from BD Biosciences, eBioscience, or Biolegend[30].

**Perfusion and brain tissue processing.** Mice were given a euthanizing dose of chloral hydrate (15%) and transcardially perfused with cold PBS followed by paraformaldehyde (4% in PBS). Brains were isolated and post-fixed overnight with paraformaldehyde. Coronal sections were cut at 50 μm thickness on a Vibratome (Leica) for immunohistochemistry (IHC) studies.

**IHC and microscopy.** Coronal slices were incubated with blocking solution (3% normal donkey serum, 0.3% Triton X-100 in PBS) for 1 h. Slices were then incubated with primary goat anti-mouse PSD95 (Abcam, 1:1000) overnight at 4°C. Slices were washed and incubated with secondary antibody for 2 h (Donkey anti-goat Cy2 or Cy5 1:400 (Jackson ImmunoReserach)). Slices were washed and stained with 1 μg/ml DAPI (Sigma) for 20 min, mounted and air-dried overnight. Slices were quickly dehydrated with various concentrations of ethanol and cover-slipped with DPX mounting medium (Electron Microscopy Sciences). Imaging of PSD95 puncta were taken on a Zeiss LSM780 and 1 μm z-stacks were taken at ×100 magnification. Deconvolution was performed on all z-stacks with AutoQuant X (Media Cybernetics).

**Assessment of IL-6 expression in vitro in PBMCs.** Whole blood from mice was mixed with complete RPMI media and laid over the Ficoll-Paque Plus (GE Healthcare, Sweden), centrifuged at 2200 r.p.m. for 15 min. The buffy coat containing PBMCs was isolated, washed once with BEP (0.5% BSA, 2 mM EDTA in PBS) and plated out in a 24-well plate at 5 × 10$^5$/well in culture medium RPMI-1640 supplemented with 20% horse serum, 10% FBS, 2 mM L-glutamine, 25 mM Hepes and 100 U/ml penicillin/streptomycin. PBMCs were treated with various phenolic acids for 16 h and challenged with 7.5 μg/ml LPS. Supernatant was collected by centrifugation following 16 h of LPS stimulation and the levels of IL-6 were measured using the Mouse IL-6 Quantikine ELISA Kit from R & D System.

**Cloning and expression of IL-6 promoter and intronic CpG-rich DNA.** CpG-rich motif from mouse IL-6 promoter and intron 1, intron 3, and intron 4 were cloned into basic (no promoter or enhancer) or promoter (minimal EF1 promoter with no enhancer) pCpG-free Lucia plasmid (InvivoGen) using In-Fusion Cloning Kit (Clontech Laboratories). Plasmids were prepared in *E. coli* GT115 cells and expressed in N2A neuoblastoma cells using lipofectamine 2000 (ThermoFisher) according to manufacturer's instruction. Following overnight recovery, cells were subjected to 5 μM of methylation inhibitor 5-Aza-2′-deoxycytidine (AZA-DC) or DHCA treatment. Twenty-four hours later, 10 μl of medium was incubated with 50 μl of QUANTI-Luc luciferase substrate for luciferase activity assessment.

**Mouse MSNs enriched primary culture and treatment.** Striatal tissue from E18 was mechanically triturated and centrifuged. Neurons were seeded onto a poly-D-lysine-coated 12-well plate at 5 × 10$^5$ cells/well and cultured in Neurobasal medium, supplemented with 2% B27, 0.5 mM L-glutamine, and 1% penicillin-streptomycin (Gibco-BRL, Invitrogen). Following 4 days in vitro (DIV), 5 μM of Ara-c was used to inhibit the growth of glial cells. Following 10-day DIV culturing, neurons were treated with select brain-bioavailable polyphenol metabolites for 16 h, washed once with cold PBS and RNA was then isolated. For virus infection, 10-day DIV cultures were infected with either HSV-EGFP or HSV-Rac1 for 48 h before RNA isolation.

**RNA isolation and gene expression assessment.** Total RNA from MSN-enriched cultures or brain NAc was isolated using the RNeasy Mini Kit (Qiagen, Valencia, CA, USA) and reverse transcribed. Gene expression was measured in four replicates by quantitative RT-PCR using Maxima SYBR Green master mix (Fermentas) in ABI Prism 7900HT. Primer sequences are listed in Supplementary Table 2. Mouse hypoxanthine phosphoribosyltransferase (HPRT) expression level was used as an internal control. Data were normalized using the 2$^{-\Delta\Delta Ct}$ method[63]. Levels of target gene mRNAs were expressed relative to those in control cultures or mice and plotted in GraphPad Prism.

**Quantitative chromatin immuneprecipitation (qCHIP).** Site-directed qChIP was performed for pan-acH3 and H3K27me3 as described previously with modification[11]. Neurons were washed once with PBS and crosslinked in 1% formaldehyde for 15 min, quenched by glycine at a final concentration of 125 mM. Neurons were then rinsed with cold PBS containing protease inhibitor cocktail (Sigma) and scraped off the plate. Cells were resuspended and sonicated with a Bioruptor (15 cycles, 30 sec on/off), which produces chromatin fragments ~350 bp in length. Two-hundred microliters of sheared chromatin was used for each IP using 2 μg of antibody to acetyl H3 or H3K27me3 (EMD Millipore) per sample in the presence of protein A/G beads at 4 °C for overnight (~16 h). Samples were washed five times with washing buffer (0.1% SDS, 1% TritonX100, 500 mM NaCl and 2 mM EDTA in 20 mM Tris-HCl, pH8) followed by one wash with LiCl washing buffer (150 mM LiCl, 1% Na-deoxycholate, 1% NP-40 and 1 mM EDTA in 10 mM Tris-HCl, pH8), and eluted by heating to 65 °C with shaking on a Thermomixer for 30 min. Chromatin was then reverse crosslinked by heating to 65 °C overnight and DNA was purified using the QIAquick PCR Purification Kit (Qiagen). Levels of specific histone modifications along the Rac1 promoter were determined by measuring the amount of acetylated or methylated histone-associated DNA using quantitative real-time PCR as previously described[11].

**Multiplex ELISA for cell signaling and mouse cytokines assays.** Luminex xMAP multiplexed immunoassays were used to evaluate the levels of phosphorylated proteins in PBMCs pre-treated with DHCA for 16 h followed by LPS stimulation. PBMCs were washed once with PBS and immediately lysed with MilliplexMAP Cell Signaling Universal Lysis Buffer and applied to the assay following manufacturer's instruction (EMD Millipore). The phosphoprotein analytes used were: AKT (Ser473), JNK (Thr183/Tyr185), ERK (Thr185/Tyr187), p38 (Thr189/Tyr182) (Millipore, MA). Multiplex MAP mouse cytokine/chemokine Panel (EMD Millipore) was used to measure the levels of 32 cytokines/chemokines in mouse plasma or PBMC cultures.

**In vitro cytotoxicity.** MTT and LDH release assays were conducted in MSN-enriched primary neurons and in PBMCs[64]. Cellular cytotoxicity was conducted in HepG2 cells using high content screening (HCS) protocols[65]. Briefly, HepG2 cells were plated on 384-well tissue culture treated black walled clear bottomed polystyrene plates and treated with varying concentrations of Mal-gluc or DHCA. The cells were loaded with the relevant dye/antibody for each cell-health marker following incubation, including cell count, markers for nuclear size, DNA structure, cell membrane permeability, mitochondrial mass, mitochondrial membrane potential and cytochrome *C* release. The plates were then scanned using an automated fluorescent cellular imager ArrayScanr (Thermo Scientific Cellomics).

**In vitro cardiotoxicity.** Cardiotoxicity was assessed using CHO cells stably transfected with the hERG using the IonWorks HT instrument (Molecular Devices Corporation), which automatically performs electrophysiology measurements in 48 single cells simultaneously in a specialized 384-well plate (PatchPlate)[42]. A

single-cell suspension was added to the wells of a PatchPlate to form an electrical seal at the bottom and pre-compound currents were recorded. Various concentrations of Mal-gluc or DHCA were added to the cells in four replicates for 5 min and post-compound currents were then recorded. Post-compound currents are then expressed as a percentage of pre-compound currents and plotted against concentration for each compound and IC50 was calculated.

**Compound stability.** Compound stability in mouse and human plasma and in mouse brain homogenate were conducted[44, 45]. Specifically, Mal-gluc or DHCA was incubated in duplicate with plasma at 37 °C and samples were taken at various time points and mixed with three volumes of ice-cold Stop Solution (methanol containing propranolol, diclofenac, or other internal standard). Samples were centrifuged to remove precipitated protein, and the supernatants were analyzed by LC/MS/MS to quantitate for remaining test compound. Data are converted to percentage remaining by dividing by the time zero concentration value. Data are fit to a first-order decay model to determine half-life. Plasma protein binding were conducted using both mouse and human plasma[45].

**Cytochrome P450 enzymes (CYP) inhibition assay.** CYP inhibition was tested in isolated microsomes for CYP2D6, CYP3A4, CYP1A2, CYP2C9, CYP2C19, CYP2C8, and CYP2B6[66]. Briefly, increasing concentrations of Mal-gluc or DHCA were incubated with human liver microsomes (HLM) in the presence of 2 mM NADPH in 100 mM potassium phosphate (pH 7.4) containing 5 mM magnesium chloride and a specific probe substrate for different CYPs: tacrine (CYP1A2), amodiaquine (CYP2C8), tolbutamide (CYP2C9), S-mephenytoin (CYP2C19), dextromethorphan (CYP2D6), midazolam/testosterone (CYP3A4/5), or bupropion (CYP2B6). CYP-specific inhibitors were included alongside the test compound as a positive control, including α-naphthoflavone (CYP1A2), quercetin (CYP2C8), ketoconazole (CYP3A4/5), quinidine (CYP2D6), sulfaphenazole (CYP2C9), and ticlopidine (CYP2C19, CYP2B6). After incubation at 37 °C, the reactions were terminated by addition of a methanol-containing internal standard (propranolol) for analytical quantification. The quenched samples were then incubated at 4 °C for 10 min and centrifuged at 4 °C for 10 min. The supernatant was removed and the probe substrate metabolite was analyzed by LC-MS/MS (Agilent Technologies Triple Quad LC-MS/MS).

**Radioligand binding assay.** Radioligand binding assays were performed as previously described[67]. Briefly, crude membrane fractions prepared from a stable HEK cell line expressing recombinant receptors were mixed with a radiolabeled specific ligand and 10 μM of competitor compound (DHCA, Mal-gluc, or non-specific ligand). The percentage of radioligand binding was subsequently measured. Compounds that showed a minimum 50% inhibition were subjected to a secondary radioligand binding assay to determine equilibrium binding.

**Overall statistics.** All values are expressed as mean and s.e.m. For dose finding, one-way ANOVAs were used to compare all groups followed by Bonferroni's comparison of testing group and the control group. For behavioral (SI and sucrose preference tests) as well as biochemical analyses comparing control group vs. testing group, one-way ANOVAs followed by Bonferroni's comparison or unpaired two-tailed student's $t$-tests with Welch's correction were used. In all studies, outliers (2 s.d. from the mean) were excluded and the null hypothesis was rejected at the 0.05 level. All statistical analyses were performed using Prism Stat program (GraphPad Software, Inc.).

**Data availability.** Data supporting the findings of this manuscript are available from the corresponding authors upon reasonable request.

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

## Acknowledgements

Funding was provided by the P50 AT008661-01 from the National Center for complementary and Integrative Health (NCCIH) and the Office of Dietary Supplements (ODS), NIH R01 MH090264, NIH R01 MH104559, NSF81200862 and support from Altschul Foundation. In addition, J.W. and G.M.P. hold positions in the Research and Development unit of the Basic and Biomedical Research and Training Program, GRECC and G.M.P. is a VA Senior Career Scientist at the James J.Peters Veterans Affairs Medical Center. We acknowledge that the contents of this manuscript are solely the responsibility of the authors and do not necessarily represent the views of the NCCIH, NIH, ODS or the U.S. Department of Veterans Affairs or the United States Government.

## Author contributions

J.W., G.E.H., H.Z., S.Z., W.Z., S.A.G., W.B., C.M., V.K., M.L., M.X., D.B., M.L.P., M.E.F., A.E.-F., S.Y. and A.S. participated in the acquisition and/or analysis of data. J.W., G.E.H., L.H., R.D., M.M., M.-H.H., S.J.R. and G.M.P participated in the design and/or interpretation of the reported experiments or results. J.W. and G.M.P wrote the manuscript and all other authors reviewed and edited the manuscript.

## Additional information

**Competing interests:** The authors declare no competing financial interests.

