## [Peer Review File · Nature Communications]

Reviewers' expertise:

Reviewer #1: neuroepigenetics;

Reviewer #2: neuro-immune interaction, inflammation, stress;

Reviewer #3: a role of the immune system in depression.

Reviewers' comments:

Reviewer #1 (Remarks to the Author):

In the manuscript by Wang and colleagues, the authors build on their previous work to explore the potential of specific phytochemicals (DHCA and Mal-gluc) to alleviate rodent behaviors akin to depressive-like symptoms. The authors first show that BDPP (bioactive dietary polyphenol preparation) treatment during repeated social defeat stress (RSDS) increases the percent of animals displaying "resilient" behavior in a subsequent social interaction task. By screening the individual components of BDPP, the authors identified DHCA and Mal-gluc to follow up on. DHCA produced a DNA methylation-dependent decrease in levels of IL-6 in PBMCs. This fits with the increase in plasma IL-6 consistently associated with depression phenotypes, including in their own data (Fig 2). Mal-gluc independently produced a histone acetylation-dependent increase in Rac1 levels in medium spiny neuron cultures. This fits with the role of nucleus accumbens Rac1 in the RSDS model of depression. After establishing doses of DHCA and Mal-gluc that produced optimal effects on IL-6 and Rac1, respectively, the compounds were combined and tested in the RSDS model by administering throughout the training period. The result was improved performance in social interaction and sucrose preference. Further, it was confirmed that IL-6 (plasma), Rac1, PSD95 (accumbens) and mEPSC frequency were normalized. The authors next performed similar experiments, but shifted DHCA/Mal-gluc treatment to after RSDS. Similar to a prescribed tricyclic antidepressant (imipramine), treatment shifted a greater percent of animals to a resilient phenotype. Bone marrow chimeras were also produced by transferring from susceptible donors to controls. Subthreshold RSDS training resulted in a susceptible phenotype in the susceptible chimeras, but not controls. DHCA and Mal-gluc were then characterized to determine their drug-like properties. Broad safety and toxicity measures (e.g. MTT, hERG), plasma and brain stability, plasma protein binding and P450s were assessed and deemed acceptable. Finally, DHCA and Mal-gluc were screened against a broad panel of receptors and transporters and no evidence was found of binding to players typically associated with depression (e.g. 5-HT receptors) at relevant drug concentrations. Overall, this is a clearly written manuscript that describes a thorough and exciting set of findings with direct therapeutic potential. Only a few minor comments:

1. A more clear transition describing the reasoning behind searching for BDPP metabolites is needed.
2. The manuscript flows well, for the most part. However, Figure 2's data is very poorly incorporated and distracting. This is particularly true of the Rac1 results. What purpose does Figure 2B serve? The current results do not seem to connect IL-6 and Rac1, yet a working model is presented that suggests this. In this reviewer's opinion, a connection is not needed and need not be forced.
3. In Figure 3A and B (left panels), it would be helpful if the authors indicated the letter that

corresponds to the compounds of interest so the reader does not have to refer back to the table.

4. The behavior timelines in Figure 5 are far too small and, while nearly impossible to read, it appears that "HDAC" is used, rather than "DHCA".

5. The number of outliers removed from specific experiments and treatment groups should be stated.

Reviewer #2 (Remarks to the Author):

In this manuscript, Wang et al introduce dietary polyphenol, and its microbiome metabolites as potential treatment for depression and stress disorders. The study uses rodent models to characterize the effects of two phytochemicals *in vivo*, *ex vivo* and *in vitro*. They provide studies to support the claim that DHCA/Mal-gluc promote resilience against stress-mediated

depression-like phenotypes. They suggest that these effects are mediated by modulation of systemic inflammatory responses and synaptic plasticity. Specifically, they demonstrate effects on IL-6 as a marker of inflammation and Rac-1 as a marker of plasticity. They propose that DHCA treatment reduces IL-6 expression by inhibiting DNA methylation at introns 1 and 3 of the IL-6 sequences. They further suggest that Mal-gluc modulates NAc synaptic plasticity by increasing histone acetylation along the promoter and upstream regions of the Rac1 gene.

Although the findings are important I have several concerns regarding the experimental designs of some of the experiments, the clarity of data presentation and the conclusions made in the study.

Below are my specific concerns:

Crucial points:

1. Generally, the authors focus only on the social defeat stress paradigm. The effects of BDPP and its metabolites needs to be validated by, at least, one additional depression paradigms such as chronic mild stress model or the learned helplessness model. In addition, the authors suggest that the compound can be used to treat depression that has chronic and lasting effects, however, all assays here were done a day or two after the RSDS.

2. RSDS model analysis: The authors apply a historical approach of data analysis. As indicated in Nat Protoc. ; 6(8): 1183–1191. "Historically, a SI ratio equal to 1, in which equal time is spent in the presence versus absence of a social target, has been used as the threshold for dividing defeated mice into the susceptible and resilient categories. Control C57BL/6J mice show a strong tendency to spend greater than or equal amounts of time in the interaction zone in each session." Therefore, this protocol suggests that: "Behavioral results from social defeat stress are reported in two ways: (i) as a comparison of total time spent by the C57BL/6J mouse in the interaction zone during each social interaction test session when the target is absent or present, or (ii) as a ratio of these two times. The social interaction ratio (SI ratio) is obtained by dividing the time spent in the interaction zone when the target is present by the time spent in the interaction zone when the target is absent."

Moreover, the authors in some cases report the result in seconds (Fig 1, 4h) and sometimes

in ratio (Fig 4c). This further highlights the question, how strong are the effects? In the image provided in Fig 1c, the effect is striking but in the analysis, the change is practically around 15 sec out of a total of 2.5 min.

3. The authors only evaluate IL-6 as a marker of inflammation. However, it is likely that after 16 hours of incubation, many factors will change in these cells, including other cytokines. Moreover, the reduction in DNMT1 expression probably affects the expression of many other genes besides IL-6. Changes in expression of other genes is important because it may eventually manifest as side effects of the treatment and are crucial for the mechanistic understanding of the process.

4. The choice of Rac-1 and IL-6 is almost arbitrary and the data provided in Fig 2 is basically a validation of previous results. Thus, it may be better for the paper to transfer that to the supporting data.

5. The authors argument that based on the chimeric mice they can conclude that the effect on IL-6 in the periphery is limited because these IL-6^{-/-} cells can also infiltrate the brain and act there to modulate activity.

6. The signaling data: the authors show that DHCA attenuates IL-6 expression, while having no effect on AP-1 and NF-κB. However, it is unclear whether the samples were treated similarly in both assays. In fact, it is unclear at what point where these molecules analyzed. It is possible that with a strong stimulation of LPS as done here, the effects may be overlooked. Also, the effect can be seen at one time point and not another, thus a time course is required. Please note that the multiplex ELISA for ERK, JNK, p38 or AKT are (page 8) is not described in the method section.

7. Fig 4D: the sucrose assay is very sensitive. Thus, prior to beginning testing, mice must be habituated to the presence of two drinking bottles for at least 3 days in their home cage. In order to detect possible side preference in the drinking behaviour, the water intake should be measured daily in both bottles. After the acclimatization, the water and sucrose solution intake should be measured daily for several days, to assure data consistency. Also, the positions of two bottles should be switched daily to reduce any confound produced by a side bias.

8. The analysis was done only in males; however, depression is more common in females. This is an important point, give the author's claim to proceed for clinical study.

9. The authors tested only one condition, yet they argue that DHCA/mal-gluc can immediately translate into human clinical studies for the treatment of stress disorders and depression either alone or in combination with currently available antidepressants.

10. Given that its the 4th publication on these compound in different brain pathologies, the novelty not clearly emphasized.

Minor comments:

1. Did the authors evaluated the anti-depressant effects of DHCA and Mal-gluc individually (not combined)?

1. Please provide statistics for Fig 3d.

2. The authors often used word such as: Greatly increased (p6), excellent safety (p23).

3. How many repeats were done for Fig 1A?

Reviewer #3 (Remarks to the Author):

Understanding the neurobiology of mood disorders and developing additional therapeutics is an integral area of study. Wang et al, authors of the manuscript, "Epigenetic modulation of inflammation and synaptic plasticity promotes resilience against stress disorder and depression," provide evidence of that specific phenolic metabolites promote behavioral resilience by modulating molecular pathways in the nucleus accumbens and peripheral immune cells. These findings are consistent with previous work, however, the authors need to provide some clarity of how neuroimmune mechanisms converge with epigenetic modulation in medium spiny neurons of the nucleus accumbens. Some experimental issues need to be addressed as well, I have outlined several points that the authors should consider prior to publication.

Points:

- 1) In Fig.2A it would be beneficial to show that PSD95 was selectively increased on medium spiny neurons and not interneurons in the nucleus accumbens. In Fig.2C the authors present a schematic to demonstrate convergent mechanisms of IL-6 and Rac1 in stress-induced depression. The authors suggest that IL-6 modulates Rac1, but provide no direct data. The authors need to show peripheral IL-6 mediates stress-induced Rac1 down-regulation, and demonstrate that Rac1 over-expression can prevent IL-6-induced depressive phenotypes.
- 2) The authors provide evidence that specific phenolic metabolites modulate LPS-induced IL-6 release from PBMCs. An important consideration is that LPS stimulation promotes significantly higher levels of IL-6 release compared to stress-related stimuli, implying different molecular pathways are engaged. Thus, it is recommended the authors provide evidence that similar modulation of IL-6 is observed in stressed PBMCs. Further circulating IL-6 in depressed/stressed individuals may be produced by sources other than leukocytes (including the liver). Do the defined metabolites modulate hepatocyte responses to stress or LPS?
- 3) In Fig.4A the authors present data that DHCA or Mal-gluc modulate IL-6 and Rac-1, respectively. Were there any metabolites that simultaneously modulated IL-6 and Rac1 expression? These findings would provide convergence of these mechanisms in observed behavioral phenotypes.
- 4) In Fig.5 the authors indicate that treatment after social stress is able to reverse social defeat deficits in sucrose consumption and splash test. The social interaction test is also presented in Fig.S5A. Though it may not be significant it would be beneficial to present with these primary findings. Further, do the authors anticipate that DHCA/Mal-Gluc treatment engages Rac1/IL-6 to reverse depressant-like behaviors after social stress? If so, they should provide results that support these assertions.

5) In Fig.5D-J the authors present results that show susceptibility to subthreshold social defeat can be “transferred” to BM-chimera mice that received susceptible donor BM. Of note, DHCA/Mal-Gluc treatment blocked this susceptibility. It appears that this may be linked to diminished stress-induced release of monocytes/granulocytes into circulation, and reduced IL-6 plasma levels. Are there baseline differences in circulating monocytes/granulocytes prior to subthreshold stress, as in are susceptible donor cells more likely to enter circulation? If not, this implies that DHCA/Mal-Gluc treatment prevents stress-induced release of these monocytes/granulocytes after subthreshold stress. In this case, the authors should assess DHCA/Mal-Gluc modulation of glucocorticoids or sympathetic nervous system mechanisms that promote egress of monocytes/granulocytes from the BM.

Other:

1) Fig.2A is lacking the IL-6^{-/-} BM chimera unstressed control. Fig.2B is performed in cell culture, this should be noted.

2) In Fig.4B the social interaction test is performed after treatment to prevent social defeat-induced depressive-like behaviors. Did DHCA/Mal-Gluc treatment alter the proportion of susceptible mice after RSDS?

3) In Fig.5H-J it is unclear what tissues are being analyzed and at what time point.

Response to Reviewers

Reviewer #1 (Remarks to the Author):

In the manuscript by Wang and colleagues, the authors build on their previous work to explore the potential of specific phytochemicals (DHCA and Mal-gluc) to alleviate rodent behaviors akin to depressive-like symptoms. The authors first show that BDPP (bioactive dietary polyphenol preparation) treatment during repeated social defeat stress (RSDS) increases the percent of animals displaying “resilient” behavior in a subsequent social interaction task. By screening the individual components of BDPP, the authors identified DHCA and Mal-gluc to follow up on. DHCA produced a DNA methylation-dependent decrease in levels of IL-6 in PBMCs. This fits with the increase in plasma IL-6 consistently associated with depression phenotypes, including in their own data (Fig 2). Mal-gluc independently produced a histone acetylation-dependent increase in Rac1 levels in medium spiny neuron cultures. This fits with the role of nucleus accumbens Rac1 in the RSDS model of depression. After establishing doses of DHCA and Mal-gluc that produced optimal effects on IL-6 and Rac1, respectively, the compounds were combined and tested in the RSDS model by administering throughout the training period. The result was improved performance in social interaction and sucrose preference. Further, it was confirmed that IL-6 (plasma), Rac1, PSD95 (accumbens) and mEPSC frequency were normalized. The authors next performed similar experiments, but shifted DHCA/Mal-gluc treatment to after RSDS. Similar to a prescribed tricyclic antidepressant (imipramine), treatment shifted a greater percent of animals to a resilient phenotype. Bone marrow chimeras were also produced by transferring from susceptible donors to controls. Subthreshold RSDS training resulted in a susceptible phenotype in the susceptible chimeras, but not controls. DHCA and Mal-gluc were then characterized to determine their drug-like properties. Broad safety and toxicity measures (e.g. MTT, hERG), plasma and brain stability, plasma protein binding and P450s were assessed and deemed acceptable. Finally, DHCA and Mal-gluc were screened against a broad panel of receptors and transporters and no evidence was found of binding to players typically associated with depression (e.g. 5-HT receptors) at relevant drug concentrations. Overall, this is a clearly written manuscript that describes a thorough and exciting set of findings with direct therapeutic potential. Only a few minor comments:

1. A more clear transition describing the reasoning behind searching for BDPP metabolites is needed.
 - *As suggested, we have revised the manuscript (pages 7 and 8) to better define the rationale behind the search for bioactive BDPP metabolites.*
2. The manuscript flows well, for the most part. However, Figure 2’s data is very poorly incorporated and distracting. This is particularly true of the Rac1 results. What purpose does Figure 2B serve? The current results do not seem to connect IL-6 and Rac1, yet a working model is presented that suggests this. In this reviewer’s opinion, a connection is not needed and need not be forced.
 - *There is currently no consensus on whether peripheral IL-6 directly influences Rac-1 in the NAc. Based on the reviewer’s recommendation, we have modified the schematics of our working hypothesis (originally presented in Fig. 2C, now Fig. S1D) by removing the dotted line connecting peripheral IL-6 with Rac1 in*

the NAc, which implicates a direct interaction between IL-6 and Rac1. The revised schematics of our working hypothesis now more accurately reflects the current knowledge that down regulation of Rac-1 in the NAc and up-regulation of IL-6 in the periphery may each contribute to depression-like behavioral phenotypes by inducing synaptic maladaptation in the NAc following RSDS. Based on the recommendation from Reviewer #2, we also transferred the revised figure to Supplementary data (Fig.S1).

3. In Figure 3A and B (left panels), it would be helpful if the authors indicated the letter that corresponds to the compounds of interest so the reader does not have to refer back to the table.

- *As suggested, we have modified Figs. 2A and 2B (previously presented as Figs. 3A and 3B) by adding the names of the corresponding compounds into each of the figures so the reader does not have to refer back to the table.*

4. The behavior timelines in Figure 5 are far too small and, while nearly impossible to read, it appears that “HDAC” is used, rather than “DHCA”.

- *We have enlarged the schematics of the “behavior timelines” in both Figs. 4A and 4F (previously presented as Figs. 5A and 5D). We have also corrected the typographic error.*

5. The number of outliers removed from specific experiments and treatment groups should be stated.

- *The number of outliers removed from each experiment is now reported in the Life Sciences Reporting Summary, Experimental design section 2 “Data exclusions”.*

Reviewer #2 (Remarks to the Author):

In this manuscript, Wang et al introduce dietary polyphenol, and its microbiome metabolites as potential treatment for depression and stress disorders. The study uses rodent models to characterize the effects of two phytochemicals in vivo, ex vivo and in vitro. They provide studies to support the claim that DHCA/Mal-gluc promotes resilience against stress-mediated depression-like phenotypes. They suggest that these effects are mediated by modulation of systemic inflammatory responses and synaptic plasticity. Specifically, they demonstrate effects on IL-6 as a marker of inflammation and Rac-1 as a marker of plasticity. They propose that DHCA treatment reduces IL-6 expression by inhibiting DNA methylation at introns 1 and 3 of the IL-6 sequences. They further suggest that Mal-gluc modulates NAc synaptic plasticity by increasing histone acetylation along the promoter and upstream regions of the Rac1 gene.

Although the findings are important I have several concerns regarding the experimental designs of some of the experiments, the clarity of data presentation and the conclusions made in the study.

Crucial points:

1. Generally, the authors focus only on the social defeat stress paradigm. The effects of BDPP and its metabolites needs to be validated by, at least, one additional depression paradigms such as chronic mild stress model or the learned helplessness model. In addition, the authors suggest that the compound can be used to treat depression that has chronic and lasting effects, however, all assays here were done a day or two after the RSDS.

- *Based on the reviewer’s recommendation, we now add new evidence, presented in Fig. 5 and described on pages 19 and 20 of the Results section, demonstrating that combined DHCA/Mal-gluc treatment is also effective in modulating depression-like behavioral phenotypes in a variable stress (VS) mouse model. Collectively, our corroborative evidence from two independent experimental paradigms – RSDS and VS - supports our conclusion that combined DHCA/Mal-gluc treatment is effective in modulating stress-mediated depression phenotypes.*

- *We now clarify in the text that for the therapeutic study, behavioral assays were conducted 2 weeks after RSDS. This is depicted in schematics of “behavior timeline” in Fig. 4A and corresponding text on page 16.*

2. RSDS model analysis: The authors apply a historical approach of data analysis. As indicated in Nat Protoc. ; 6(8): 1183–1191. “Historically, a SI ratio equal to 1, in which equal time is spent in the presence versus absence of a social target, has been used as the threshold for dividing defeated mice into the susceptible and resilient categories. Control C57BL/6J mice show a strong tendency to spend greater than or equal amounts of time in the interaction zone in each session.” Therefore, this protocol suggests that: “Behavioral results from social defeat stress are reported in two ways: (i) as a comparison of total time spent by the C57BL/6J mouse in the interaction zone during each social interaction test session when the target is absent or present, or (ii) as a ratio of these two times. The social interaction ratio (SI ratio) is obtained by dividing the time spent in the interaction zone when the target is present by the time spent in the interaction zone when the target is absent.” Moreover, the authors in some cases report the result in seconds (Fig 1, 4h) and sometimes in ratio (Fig 4c). This further highlights the question, how strong are the effects? In the image provided in Fig 1c, the effect is striking but in the analysis, the change is practically around 15 sec out of a total of 2.5 min.

- *This is an important point. For consistency and to avoid confusion, in the revised manuscript we only show social interaction ratio (SI ratio) as the measure of social interaction across all studies.*
- *Based on our experience, animals without RSDS usually spend between 80-100 seconds out of a total of 2.5 minutes testing time in the interacting zone. In comparison, our data shows that RSDS mice spent ~62 seconds in the interacting zone, whereas RSDS mice with BDPP treatment spent ~75 seconds, which approaches the normal range of time that animals without RSDS spend in the interacting zone. Therefore, we can confidently conclude that the treatment is effective. As suggested we now modify the heatmap in Fig.1C to more accurately reflect the effect of BDPP.*

3. The authors only evaluate IL-6 as a marker of inflammation. However, it is likely that after 16 hours of incubation, many factors will change in these cells, including other cytokines. Moreover, the reduction in DNMT1 expression probably affects the expression of many other genes besides IL-6. Changes in the expression of other genes are important because it may eventually manifest as side effects of the treatment and are crucial for the mechanistic understanding of the process.

- *We agree with the reviewer that after 16 hours LPS stimulation may also change the expression of many other factors by PBMCs. We now add new data (Supplementary Fig. S3. and corresponding text on pages 10 and 11) showing that in addition to IL-6, treatment of PBMCs with LPS also induced expression of other cytokines. Moreover, we demonstrate that LPS-mediated induction of a number of cytokines, particularly inflammatory cytokines, are significantly reduced by DHCA treatment. In control studies, we confirm that DHCA treatment without LPS stimulation has no detectable effect on cytokine production.*

4. The choice of Rac-1 and IL-6 is almost arbitrary and the data provided in Fig 2 is basically a validation of previous results. Thus, it may be better for the paper to transfer that to the supporting data.

- *As suggested by the reviewer, we now transfer Fig. 2 to supplementary Fig. S1.*

5. The authors’ argument that based on the chimeric mice they can conclude that the effect on IL-6 is in the periphery is limited because these IL-6^{-/-} cells can also infiltrate the brain and act there to modulate activity.

- *We agree with the reviewer that the effect of IL-6 in RSDS mice is not necessarily limited to the periphery, as a recent report (Menard et al., Nat. Neurosci. 2017) demonstrated that social defeat stress induces neurovascular pathology and increases blood brain barrier permeability which may facilitate infiltration of larger molecules and immune cells into the CNS. We now revise the text to include the potential effect of brain infiltrated IL-6 producing cells in modulating brain activity. The revised text on page 25 now reads: “Our evidence demonstrated a cause-effect relationship among leukocyte-derived pro-inflammatory responses, brain reward circuitry synaptic remodeling and the manifestation of*

depression-like behavioral phenotypes, which supports the consideration of IL-6 and IL-6 producing cells (and perhaps additional pro-inflammatory molecules), as a key therapeutic target for treating depression. This is further supported by recent findings that social defeat stress induces neurovascular pathology and increases blood brain barrier permeability that may facilitate larger molecules such as IL-6 and possibly immune cells infiltration into the CNS”

6. The signaling data: the authors show that DHCA attenuates IL-6 expression, while having no effect on AP-1 and NF- κ B. However, it is unclear whether the samples were treated similarly in both assays. In fact, it is unclear at what point where these molecules analyzed. It is possible that with a strong simulation of LPS as done here, the effects may be overlooked. Also, the effect can be seen at one time point and not another, thus a time course is required. Please note that the multiplex ELISA for ERK, JNK, p38 or AKT are (page 8) is not described in the method section.

- *We now clarify on page 9 of the “Results” section that all samples used for the signaling pathway study were treated similarly as samples used for the IL-6 screening study.*
- *Based on the reviewer’s recommendation, we have also analyzed changes of signaling molecules over time and resultant data is now presented in supplementary Fig. S2 (previously Fig. S1) with corresponding text on page 9. We also added the method for the multiplex ELISA on page 36.*

7. Fig 4D: the sucrose assay is very sensitive. Thus, prior to beginning testing, mice must be habituated to the presence of two drinking bottles for at least 3 days in their home cage. In order to detect possible side preference in the drinking behavior, the water intake should be measured daily in both bottles. After the acclimatization, the water and sucrose solution intake should be measured daily for several days, to assure data consistency. Also, the positions of two bottles should be switched daily to reduce any confound produced by a side bias.

- *We have modified the Methods section on page 29 to clarify that mice were treated with either vehicle or individual test compounds for 14 days. Throughout this period, mice were presented with two drinking bottles to acclimate the animals such that when we conducted the sucrose preference test at the end of the study, the animals had been accustomed to the presence of the two drinking bottles. We also regularly switch the two bottles. In the revised results, we now also included the sucrose preference test measured at 48 hours and 72 hours in Fig. 3D (previously Fig. 4D) with corresponding text on page 13.*

8. The analysis was done only in males; however, depression is more common in females. This is an important point, give the author’s claim to proceed for clinical study.

- *As suggested by the reviewer, we now provide new information generated using the subchronic variable stress model (SCVS) in female mice. We report that DHCA/Mal-gluc treatment is also effective in improving variable stress-induced depression-like behavior of self-neglect and anxiety in females (Fig. 5 and corresponding texts on pages 20). In the Discussion of the revised manuscript (page 26) we acknowledge that depression is more prevalent among females and our data demonstrate that the treatment is effective for both male and female mice.*

9. The authors tested only one condition, yet they argue that DHCA/mal-gluc can immediately translate into human clinical studies for the treatment of stress disorders and depression either alone or in combination with currently available antidepressants.

- *We now include new evidence in Fig. 5 and corresponding results on pages 19 and 20 demonstrating the efficacy of DHCA/Mal-gluc to modulate depression behavioral phenotypes in two independent models of stress-mediated depression phenotypes – the RSDS and the VS model. This evidence, together with the new evidence that DHCA/mal-gluc is also effective in female model of depression, confirmed the efficacy of DHCA/Mal-gluc treatment to modulate depression-like behavioral phenotypes.*

10. Given that its the 4th publication on these compound in different brain pathologies, the novelty not clearly emphasized.

- *We now revise the manuscript to further emphasize the novelty of developing DHCA/Mal-gluc for depression/anxiety in the Discussion on page 23, second paragraph. We now state that “the effect of BDPP on depression was never before tested. Here we demonstrated that oral administration of BDPP is effective in attenuating the development of depression phenotype in a well-established RSDS model in mice. Moreover, we identified Mal-gluc and DHCA, two bioavailable metabolites derived from xenobiotic metabolism and gut microbiome metabolism of BDPP, can both prophylactically prevent and therapeutically treat RSDS-induced depression phenotypes, such as social avoidance and anhedonia, through epigenetic modulation of peripheral IL-6 and Rac1 in the NAc”.*

Minor comments:

1. Did the authors evaluated the anti-depressant effects of DHCA and Mal-gluc individually (not combined)?

- *We did evaluate the effect of DHCA and Mal-gluc individually and found that neither was effective in modulating RSDS-induced depression phenotype when administered alone. These results are presented in Supplementary Fig. S7 with corresponding text on page 15.*

1. Please provide statistics for Fig 3d.

- *We have clarified the statistics for both the left panel and the right panel of Fig. 2D (previously Fig. 3D) in the figure legend.*

2. The authors often used word such as: Greatly increased (p6), excellent safety (p23).

- *As suggested, we have removed these words.*

3. How many repeats were done for Fig 1A?

- *We have done two repetitions for Fig. 1A study and we have included this information in the Life Sciences Reporting Summary Form, Experimental design section 3 “Replication”.*

Reviewer #3 (Remarks to the Author):

Understanding the neurobiology of mood disorders and developing additional therapeutics is an integral area of study. Wang et al, authors of the manuscript, “Epigenetic modulation of inflammation and synaptic plasticity promotes resilience against stress disorder and depression,” provide evidence of that specific phenolic metabolites promote behavioral resilience by modulating molecular pathways in the nucleus accumbens and peripheral immune cells. These findings are consistent with previous work, however, the authors need to provide some clarity of how neuroimmune mechanisms converge with epigenetic modulation in medium spiny neurons of the nucleus accumbens. Some experimental issues need to be addressed as well, I have outlined several points that the authors should consider prior to publication.

Points:

1) In Fig.2A it would be beneficial to show that PSD95 was selectively increased on medium spiny neurons and not interneurons in the nucleus accumbens. In Fig.2C the authors present a schematic to demonstrate convergent mechanisms of IL-6 and Rac1 in stress-induced depression. The authors suggest that IL-6 modulates Rac1, but provide no direct data. The authors need to show peripheral IL-6 mediates stress-induced Rac1 down-regulation, and demonstrate that Rac1 over-expression can prevent IL-6-induced depressive phenotypes.

- *Our working hypothesis is that peripheral IL-6 may contribute to stress-induced depression phenotype by influencing synaptic plasticity in the NAc. This is supported by our evidence that in comparison to wild-type chimera mice, depletion of peripheral IL-6 in IL-6^{-/-}BM chimera mice results in significantly reduced*

PSD95, a marker of excitatory synapses, in the NAc following RSDS. Based on the reviewer's recommendation, we conducted a series of studies to investigate, quantitatively, whether induction of PSD95 in the NAc following RSDS is specific to medium spiny neurons. Due to technical difficulties, we were not able to conclusively determine the elevated PSD95 is selective to the medium spiny neurons and not the interneurons. However, this does not interfere with our working hypothesis nor the interpretation of the conclusion of our studies. Nonetheless we now state in the result on page 6-7 of the revised manuscript "Whether stress-induced IL-6-mediated up-regulation of PSD95 in the NAc is specific to select cell types (e.g., Drd1 or Drd2 medium spiny neurons, or inter-neurons) or is non-specifically induced across multiple cell types in the NAc needs further characterization in the future."

- *We agree with the reviewer's critique regarding the schematic representation of our working hypothesis presented in Fig. S1D (Previously Fig.2C). We also noted that Reviewer #1 also criticized the presence of a dotted line in the schematic connecting peripheral IL-6 with Rac1 in the NAc, implicating an unfound direct interaction between IL-6 and Rac1. The revised schematic of our working hypothesis (Fig. S1D of the revised manuscript) now more accurately reflects the current consensus that down regulation of Rac-1 in the NAc and up-regulation of IL-6 in the periphery following RSDS may each contribute to synaptic maladaptation in the NAc that leads to depression-like behavioral phenotypes.*

2) The authors provide evidence that specific phenolic metabolites modulate LPS-induced IL-6 release from PBMCs. An important consideration is that LPS stimulation promotes significantly higher levels of IL-6 release compared to stress-related stimuli, implying different molecular pathways are engaged. Thus, it is recommended the authors provide evidence that similar modulation of IL-6 is observed in stressed PBMCs. Further circulating IL-6 in depressed/stressed individuals may be produced by sources other than leukocytes (including the liver). Do the defined metabolites modulate hepatocyte responses to stress or LPS?

- *Our data demonstrates that DHCA down-regulates LPS-induced IL-6 expression in primary PBMC culture, in part, through down regulation of the epigenetic modifier DNMT1 (Fig. 2C). We now provide evidence demonstrating that following RSDS, circulating PBMCs from vehicle-treated mice showed significantly higher level of IL-6 mRNA and increased DNMT1 mRNA in the PBMCs compared to non-RSDS mice. Treatment with DHCA/Mal-gluc significantly reduced RSDS-mediated induction of IL-6 mRNA as well as reduced DNMT1 mRNA expression in circulating PBMCs (see Fig. S6 and corresponding text on page 14). We acknowledge the possibility that LPS-mediated induction of IL-6 in cultured PBMCs might be mechanistically different from RSDS-mediated induction of circulating IL-6. However, our data suggests that DHCA reduces IL-6 through down regulation of DNMT1 mRNAs both in in vitro PBMC cultures and in circulating PBMCs following RSDS.*
- *The reviewer pointed out that stress-induced circulating IL-6 could be derived from PBMCs as well as other tissues such as hepatocytes. In our study, we found that compared to the WT BM transplanted chimeras, IL-6^{-/-} BM transplanted chimeras had a significantly lower level of circulating IL-6 following RSDS (Fig. S1B and text on page 7). The level is comparable to non-stressed WT BM mice or non-stressed IL-6^{-/-} BM chimera (Fig. S1B), suggesting that blood cells are the main source of circulating IL-6 following RSDS. However, it is possible that alternative sources may also contribute to stress-induced circulating IL-6. This possibility is now stated in the text on page 7.*

3) In Fig.4A the authors present data that DHCA or Mal-gluc modulates IL-6 and Rac-1, respectively. Were there any metabolites that simultaneously modulated IL-6 and Rac1 expression? These findings would provide convergence of these mechanisms in observed behavioral phenotypes.

- *The original screening was performed with the intention to find select bioavailable polyphenol metabolites that can simultaneously modulate IL-6 and Rac1. However the high throughput screening of the currently available polyphenol metabolites did not identify any metabolites capable of simultaneously down-regulating IL-6 and up-regulating Rac-1 (Figs 2A and 2B). This is clarified on page 8.*

4) In Fig.5 the authors indicate that treatment after social stress is able to reverse social defeat deficits in

sucrose consumption and splash test. The social interaction test is also presented in Fig.S5A. Though it may not be significant it would be beneficial to present with these primary findings. Further, do the authors anticipate that DHCA/Mal-gluc treatment engages Rac1/IL-6 to reverse depressant-like behaviors after social stress? If so, they should provide results that support these assertions.

- *As suggested by the reviewer, we now present the SI data for the therapeutic study in Fig 4B.*
- *Our hypothesis is that the post-stress treatment with DHCA/Mal-gluc will normalize Rac1/IL-6 to produce therapeutic effects. We found that the treatment of susceptible mice engages Rac1 in the NAc. This is reflected by increased expression of Rac1 in the NAc following the treatment. However, we found that in both groups, the circulating IL-6 was back to baseline levels two weeks after the defeat regardless of treatment status. This is largely consistent with our previous studies (Hodes et al., Proc. Natl. Acad. Sci. U. S. A., 2014). These findings are now presented in Fig. 4E with corresponding text on page 16.*

5) In Fig.5D-J the authors present results that show susceptibility to subthreshold social defeat can be “transferred” to BM-chimera mice that received susceptible donor BM. Of note, DHCA/Mal-gluc treatment blocked this susceptibility. It appears that this may be linked to diminished stress-induced release of monocytes/granulocytes into circulation, and reduced IL-6 plasma levels. Are there baseline differences in circulating monocytes/granulocytes prior to subthreshold stress, as in are susceptible donor cells more likely to enter circulation? If not, this implies that DHCA/Mal-gluc treatment prevents stress-induced release of these monocytes/granulocytes after subthreshold stress. In this case, the authors should assess DHCA/Mal-gluc modulation of glucocorticoids or sympathetic nervous system mechanisms that promote egress of monocytes/granulocytes from the BM.

- *Prior to microdefeat, mice receiving BM from susceptible donor showed no differences in the frequency of monocytes or neutrophils compared to mice receiving BM from naïve donor (Fig. S9) suggesting susceptibility of donor mice has no significant impact on the reconstitution of blood cells in periphery. Previous studies by Heidt et. al. have shown that both in humans and in rodents, chronic stress induces monocytosis and neutrophilia, and these responses are mediated, in part, by increased activation of the sympathetic nervous system, specifically involving the β 3 adrenergic receptor (Heidt et al., Nat. Med. 2014). However, we found that neither DHCA nor Mal-gluc can effectively interact with β 3 or other adrenergic receptors (Table 2), indicating that the sympathetic nervous system is unlikely to be involved in DHCA/Mal-gluc mediated reduction of stress-induced moncytosis and nerutrophilia. In the revised Figs. 4M and 4N and on page 18, we now show that 24 hours after the subthreshold defeat, there were significantly higher levels of granulocyte-colony stimulating factor (G-CSF) and granulocyte macrophage-colony stimulating factor (GM-CSF) in the plasma from susceptible BM chimeras compared to the control BM chimeras. G-CSF plays an important role in the proliferation of neutrophils and GM-CSF plays important role in the proliferation of granulocytes and monocytes in the BM. Our data suggests that DHCA/Mal-gluc treatment attenuates stress-induced expression of G-CSF and GM-CSF, which may contribute to the reduced proliferation and release of neutrophils and monocytes from bone marrow. We also discuss the role of sympathetic nervous system on monocytosis and neurtrophilia on page 26.*

Other:

1) Fig.2A is lacking the IL-6^{-/-} BM chimera unstressed control. Fig.2B is performed in cell culture, this should be noted.

- *In the revised Fig. S1A (previously Fig. 2A), we now provide the quantification of PSD95 and associated image for the IL6^{-/-} unstressed chimera control.*
- *We now clarify on page 7 and in Figure legend that Fig. S1C (previously Fig. 2B) is performed in medium spiny neuron enriched primary culture.*

2) In Fig.4B the social interaction test is performed after treatment to prevent social defeat-induced depressive-like behaviors. Did DHCA/Mal-gluc treatment alter the proportion of susceptible mice after RSDS?

- *DHCA/Mal-gluc treatment significantly reduced the proportion of susceptible mice after RSDS. In Fig. 3C (previously fig. 4C) we now provide the scatter plot showing that about 50% of the vehicle-treated RSDS mice were susceptible while only 19% of the DHCA/Mal-gluc-treated RSDS mice were susceptible.*

3) In Fig.5H-J it is unclear what tissues are being analyzed and at what time point.

- *On page 18, we now clarify that the data presented in Figs. 4J-L (previously Figs.5H-J) are leukocytes isolated from BM transplanted mice after behavioral testing (scheme in Fig. 4F). The leukocytes were analyzed by flow cytometry to determine the chimerism and cell composition (Please see Page 33 for method).*

REVIEWERS' COMMENTS:

Reviewer #2 (Remarks to the Author):

I would like to thank the authors for carefully addressing all my concerns. The only point I would like to highlight is the identification of the various cytokines affected in the treatment (presented in Fig S3). These are very interesting results that, in my opinion, require a more elaborate discussion in the text.

Reviewer #3 (Remarks to the Author):

The authors have responded to my concerns and provided additional data to support their conclusions. While the primary concern about convergence of epigenetic (Rac1) and neuroimmune (IL-6) mechanisms in RSDS was not entirely reconciled, the authors have altered interpretation of these findings to suggest these mechanisms act independently. Additional studies to address reviewer concerns raise compelling questions about mechanisms by which DHA/Mal-gluc regulates CVS stress responses and modulates hematopoiesis in RSDS. However, with the extent of the present work further studies are not within the scope of this manuscript.

Reviewer #2 (Remarks to the Author):

I would like to thank the authors for carefully addressing all my concerns.

The only point I would like to highlight is the identification of the various cytokines affected in the treatment (presented in Fig S3). These are very interesting results that, in my opinion, require a more elaborate discussion in the text.

- *We agree with the reviewer that the various cytokines modulated by DHCA treatment, particularly the cytokines of pro-inflammation nature are very interesting. In the revised Discussion on page 17, second paragraph, we now state “Notably, besides IL-6, DHCA is also capable of modulating other inflammatory cytokines including IL-1 β and IL-12, both of which have been reported to be elevated in MDD subjects. Whether these changes are the direct effect of DHCA or secondary to IL-6 induction, and their potential contribution to attenuate stress-induced depression need further investigation.”*